# AN OPTIMAL DISCRIMINATOR WEIGHTED IMITATION PERSPECTIVE FOR REINFORCEMENT LEARNING

**Haoran Xu**[1*]**, Shuozhe Li**[1*]**, Harshit Sikchi**[1]**, Scott Niekum**[2]**, Amy Zhang**[1,3]
[1] University of Texas at Austin, [2] UMass Amherst, [3] Meta AI

## ABSTRACT

We introduce Iterative Dual Reinforcement Learning (IDRL), a new method that takes an optimal discriminator-weighted imitation view of solving RL. Our method is motivated by a simple experiment in which we find training a discriminator using the offline dataset plus an additional expert dataset and then performing discriminator-weighted behavior cloning gives strong results on various types of datasets. That optimal discriminator weight is quite similar to the learned visitation distribution ratio in Dual-RL, however, we find that current Dual-RL methods do not correctly estimate that ratio. In IDRL, we propose a correction method to iteratively approach the optimal visitation distribution ratio in the offline dataset given no addtional expert dataset. During each iteration, IDRL removes zero-weight suboptimal transitions using the learned ratio from the previous iteration and runs Dual-RL on the remaining subdataset. This can be seen as replacing the behavior visitation distribution with the optimized visitation distribution from the previous iteration, which theoretically gives a curriculum of improved visitation distribution ratios that are closer to the optimal discriminator weight. We verify the effectiveness of IDRL on various kinds of offline datasets, including D4RL datasets and more realistic corrupted demonstrations. IDRL beats strong Primal-RL and Dual-RL baselines in terms of both performance and stability, on all datasets. Project page at `https://ryanxhr.github.io/IDRL/`.

## 1  INTRODUCTION

Offline reinforcement learning (RL) has attracted attention as it enables learning policies by utilizing only pre-collected data. It is a promising area for bringing RL into real-world domains, such as industrial control (Zhan et al., 2022) and robotics (Kalashnikov et al., 2021). In such scenarios, arbitrary exploration with untrained policies is costly or dangerous, but sufficient prior data is available. The major challenge of offline RL is the distributional shift issue (Levine et al., 2020), i.e., the optimized policy's distribution is different from the offline data distribution, which means the value function may have extrapolation errors to unseen actions produced by the optimized policy (Fujimoto et al., 2018). To deal with this challenge, most offline RL algorithms build on the *primal* form of RL (i.e., maximizing a value function respect to the policy, which needs to alternate between policy evaluation and policy improvement), and impose an additional behavior constraint either by policy constraint, which constrains the policy to be close to the behavior policy using some distance function (Wu et al., 2019; Fujimoto & Gu, 2021); by value regularization, which directly modifies the value function to be pessimistic (Kumar et al., 2020; Kostrikov et al., 2021a); or by uncentainty estimation, which guides policy optimization to low-uncertainty regions (An et al., 2021; Yu et al., 2020). However, in *Primal-RL* based methods, the policy may still output out-of-distribution (OOD) actions if the policy constraint weight is not set properly, which will cause either inaccurate or over-constrained value estimation that results in suboptimal behavior.

Notably, recently there's also another line of in-sample based offline RL algorithms (Xu et al., 2023; Mao et al., 2024a;b), which can be derived from the *dual* formulation of RL (i.e., maximizing the reward respect to the policy's visitation distribution) (Sikchi et al., 2023b). Different from the primal counterpart, *Dual-RL* based methods isolate the learning of value function and policy; the value function can be learned using only dataset actions, which brings minimal estimation errors. To extract

---

*Equal contribution. Correspondence to Haoran Xu.

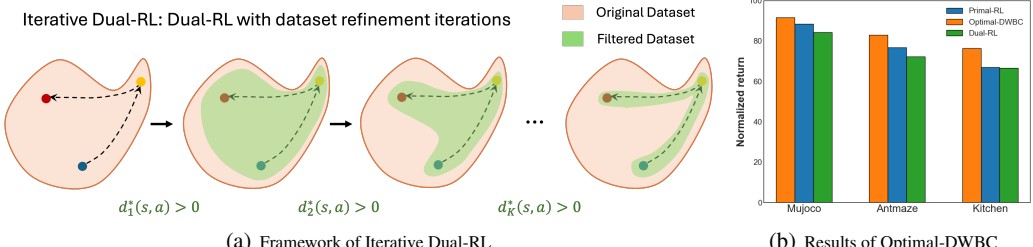

(a) Framework of Iterative Dual-RL

(b) Results of Optimal-DWBC

Figure 1: (a) Illustration of our proposed IDRL framework. IDRL breaks the regularization barrier by performing imitation learning on a iteratively-refined dataset, it can solve hard tasks where previous behavior-regularized offline RL can not do. For example, previous methods will fail at finding the shortest path from blue point to red point while crossing yellow point, due to non-uniform data coverage at different state. (b) Mean scores of optimal discriminator-weighted behavior cloning (Optimal-DWBC) on D4RL Mujoco-{m,m-r,m-e}, Antmaze-{all} and Kitchen-{all} datasets. We train a discriminator $d$ on offline dataset $\mathcal{D}$ and an additional expert dataset $\mathcal{D}_E$, we then use $w_d(s,a) = \frac{d^E(s,a)}{d^{\mathcal{D}}(s,a)} = \frac{d(s,a)}{1-d(s,a)}$ to do weighted-BC on $\mathcal{D}$ if $w_d(s,a) > \delta$. We compare it with SOTA *Primal-RL* method ReBRAC (Tarasov et al., 2024) and SOTA *Dual-RL* method ODICE (Mao et al., 2024a).

the implicit policy contained in the value function, one often uses the visitation distribution ratio between the optimized and behavior policy to do weighted behavior cloning (BC) (Peng et al., 2019). Although *Dual-RL* methods suffer minimal OOD errors by using in-sample learning, they often lag behind in performance of primal-RL methods. One conventional opinion is using weighted-BC style policy extraction causes poor generalization combined with its mode-covering nature (Park et al., 2024). In this paper, we aim to investigate pitfalls and push the limit of *Dual-RL* methods. **We start from a surprising finding: learning the optimal visitation distribution ratio using a "gold" discriminator on the offline dataset and an additional expert dataset, and using this ratio as a filtering weight in weighted-BC outperforms all classical offline RL methods on various datasets, shown in Figure 1**. Although this discriminator-weighted BC is an oracle baseline as the expert datasets are unavailable, it points to the fact that *Dual-RL* methods, as a virtue of learning visitation distribution ratios, could faciliate this kind of dataset filtering and weighting.

Given this potential, what is missing in current *Dual-RL* methods? 1) We show that current *Dual-RL* methods are unable to accurately estimate the state-action visitation ratio as a result of gradient update strategies they use for minimizing the Bellman residual. We find using a semi-gradient update in *Dual-RL* changes the fixed-point of optimization, learning an action distribution ratio rather than a visitation distribution ratio. Using the action distribution ratio to learn policies forces learning potentially suboptimal actions at states never visited by an expert, leading to poor generalization during weighted-BC policy extraction. 2) Irrespective of the gradient update strategy used, *Dual-RL* methods learn a regularized optimal visitation distribution w.r.t offline data distribution (i.e., visitation that maximizes performance while staying close to offline data), as opposed to learning the optimal visitation distribution ratio. Our key contribution in this paper is to address both the limitations above and present a new offline RL algorithm that is able to break the regularization barrier. First, we propose a new objective for accurately learning the visitation distribution ratio using a two-stage procedure. After getting the action distribution ratio using semi-gradient *Dual-RL*, we adopt an off-policy evaluation approach that recovers the correct state-action visitation distribution ratio using the action distribution ratio. This allows decomposing the original tasks of obtaining visitation distribution ratio into two distinct, complementary, and easy to solve tasks. Second, we show that the regularized optimal visitation ratio has a sparse form (some weights are zero) (Xu et al., 2023). This property can be used to refine the offline dataset iteratively and gradually learn an optimal visitation ratio that moves closer to the optimal discriminator weight and can be used for learning a better policy by weighted behavior cloning. Our theoretical results justify our method by deriving a smaller optimality lower bound due to this curriculum-refined dataset filtering.

We term our method *iterative Dual-RL* (IDRL), IDRL iteratively filters out suboptimal transitions and then performs imitation learning on the remaining subdataset that is within or close to the optimal visitation distribution. An overview of our method can be found in Figure 1. This learning paradigm also has connections to some recent success in using deep generative models (DGMs) in offline RL algorithms (Mao et al., 2024b; Hansen-Estruch et al., 2023). These methods first fit the complex, multi-modal behavior distribution using the strong expressivity of DGMs, and then

use conditioned or guided generation to unearth optimal actions. Compared to these generative methods, our method bypasses the costly two-step procedure and uses lightweight behavior cloning to directly fit the "optimal" distribution. Empirically, we verify the effectiveness of IDRL on different kinds of offline datasets, including D4RL benchmark datasets and corrupted demonstrations that is heteroskedastic and occur more often in real-world scenarios. On D4RL datasets, IDRL matches or surpasses both *Primal-RL* and *Dual-RL* baselines. On corrupted demonstrations, IDRL outperforms strong support-based and reweighting-based approaches. Under all settings, IDRL achieves better stability and robustness during inference time compared to all other baselines.

## 2 PRELIMINARIES

We consider the RL problem presented as a Markov Decision Process (Sutton et al., 1998), which is specified by a tuple $\mathcal{M} = \langle \mathcal{S}, \mathcal{A}, \mathcal{P}, d_0, r, \gamma \rangle$. Here $\mathcal{S}$ and $\mathcal{A}$ are state and action space, $\mathcal{P}(s'|s, a)$ and $d_0$ denote transition dynamics and initial state distribution, $r(s, a)$ and $\gamma$ represent reward function and discount factor, respectively. The goal of RL is to find a policy $\pi(a|s)$ which maximizes expected return $J(\pi) = \mathbb{E}[\sum_{t=0}^{\infty} \gamma^t \cdot r(s_t, a_t)]$. Offline RL considers the setting where interaction with the environment is prohibited, and one needs to learn the optimal $\pi$ from a static dataset $\mathcal{D} = \{s_i, a_i, r_i, s_i'\}_{i=1}^{N}$. We denote the empirical behavior policy of $\mathcal{D}$ as $\mu$, which represents the conditional distribution $p(a|s)$ observed in the dataset.

**Value functions and visitation distributions** Let $V^\pi : \mathcal{S} \to \mathbb{R}$ and $Q^\pi : \mathcal{S} \times \mathcal{A} \to \mathbb{R}$ be the state and state-action value function of $\pi$, where $V^\pi(s) = \mathbb{E}_\pi \left[ \sum_{t=0}^{\infty} \gamma^t r(s_t, a_t)|s_0 = s \right]$ and $Q^\pi(s, a) = \mathbb{E}_\pi \left[ \sum_{t=0}^{\infty} \gamma^t r(s_t, a_t)|s_0 = s, a_0 = a \right]$. The visitation distribution $d^\pi$ is defined as $d^\pi(s, a) = (1 - \gamma) \sum_{t=0}^{\infty} \gamma^t \Pr(s_t = s, a_t = a \mid s_0 \sim d_0, \forall t, a_t \sim \pi(s_t), s_{t+1} \sim \mathcal{P}(s_t, a_t))$, which measures how likely $\pi$ is to encounter $s, a$ when interacting with $\mathcal{M}$, averaging over time via $\gamma$-discounting. Let $V^*$, $Q^*$ and $d^*$ denote the value functions and visitation distribution corresponding to the regularized optimal policy $\pi^*$. We denote $d^E$ as the visitation distribution of the true optimal policy $\pi^E$. We denote the empirical visitation distribution of $\mu$ as $d^\mathcal{D}$. Let $\mathcal{T}_r^\pi$ be the Bellman operator with policy $\pi$ and reward $r$ such that $\mathcal{T}_r^\pi Q(s, a) = r(s, a) + \gamma \mathbb{E}_{s' \sim \mathcal{P}(\cdot|s, a), a' \sim \pi(\cdot|s')} [Q(s', a')]$. We also define two Bellman operators $\mathcal{T}_r$ and $\mathcal{T}$ for the state value function, where $\mathcal{T}_r V(s, a) = r(s, a) + \gamma \mathbb{E}_{s' \sim \mathcal{P}(\cdot|s, a)} [V(s')]$ and $\mathcal{T} V(s, a) = \gamma \mathbb{E}_{s' \sim \mathcal{P}(\cdot|s, a)} [V(s')]$.

### 2.1 PRIMAL AND DUAL RL

Interestingly, the value of a policy, $J(\pi)$, may be expressed in two ways, where the primal form uses the value function while the dual form uses the visitation distribution:

$$\boxed{Primal\text{-}RL} \quad (1 - \gamma) \cdot \mathbb{E}_{s0 \sim \mu_0, a_0 \sim \pi}[Q^\pi(s_0, a_0)] = J(\pi) = \mathbb{E}_{(s,a) \sim d^\pi}[r(s, a)]. \quad \boxed{Dual\text{-}RL}$$

In *Primal-RL*, there are typically two steps, policy evaluation and policy improvement. In policy evaluation, the $Q$-values are estimated by finding the fixed point of the Bellman recurrence (Watkins & Dayan, 1992; Sutton et al., 2008; Mnih et al., 2015), i.e., minimizing the squared single-step Bellman difference using off-policy data $\mathcal{D}$ as $\min_Q J(Q) = \frac{1}{2} \mathbb{E}_{(s,a) \sim \mathcal{D}} \left[ ((\mathcal{T}_r^\pi Q - Q)(s, a)^2 \right]$. In policy improvement, the policy $\pi$ can be optimized by $\max_\pi \mathbb{E}_{s \sim \mathcal{D}, a \sim \pi(s)} [Q(s, a)]$ using policy gradient (Sutton et al., 1999). In practice, the policy evaluation and improvement step are alternated till convergence to the optimal solution $Q^*$ and $\pi^*$. In the offline setting, to prevent overestimation arising due to the distribution shift issue, one often imposes a *policy* constraint (action-level) to the policy improvement step (Kumar et al., 2019; Fujimoto & Gu, 2021); to the policy evaluation step (Kumar et al., 2020; An et al., 2021); or to both steps (Fakoor et al., 2021; Tarasov et al., 2024).

*Dual-RL*, also known as Distribution Correction Estimation (DICE) (Nachum & Dai, 2020; Sikchi et al., 2023b), was first used to ensure unbiased estimation of the on-policy policy gradient using off-policy data. *Dual-RL* incorporates $\mathcal{J}(\pi)$ with a *visitation* constraint term (state-action level) $D_f(d^\pi \| d^\mathcal{D}) = \mathbb{E}_{(s,a) \sim d^\mathcal{D}}[f(\frac{d^\pi(s,a)}{d^\mathcal{D}(s,a)})]$ where $f(x)$ is a convex function, i.e., finding $\pi^*$ satisfying

$$\pi^* = \arg \max_\pi \mathbb{E}_{(s,a) \sim d^\pi}[r(s, a)] - \alpha D_f(d^\pi \| d^\mathcal{D}). \quad (1)$$

This regularized learning objective is generally intractable due to the dependency on $d^\pi(s, a)$, especially under the offline setting. However, by imposing the Bellman-flow constraint (Manne,

1960), $\sum_{a \in \mathcal{A}} d(s,a) = (1-\gamma)d_0(s) + \gamma \sum_{\tilde{s},\tilde{a}} d(\tilde{s},\tilde{a})\mathcal{P}(s|\tilde{s},\tilde{a})$ on states and applying Lagrangian duality and convex conjugate, its dual problem has the following tractable unconstrained form:

$$\min_V (1-\gamma)\mathbb{E}_{s \sim d_0}[V(s)] + \alpha\mathbb{E}_{(s,a) \sim d^{\mathcal{D}}}\left[f_p^*\left([\mathcal{T}_r V(s,a) - V(s)]/\alpha\right)\right], \tag{2}$$

where $f_p^* = \max\left(0, f'^{-1}(x)\right)(x) - f\left(\max\left(0, f'^{-1}(x)\right)\right)$. Note that objective (2) can be calculated solely with $(s,a,s')$ sample from $\mathcal{D}$, which enables in-sample learning and mitigating the distribution shift issue in offline RL by not querying function-approximators on out-of-distribution actions. In principle, *Dual-RL* serves as a better off-policy (by getting unbiased on-policy policy gradient estimation) and offline algorithm (no OOD actions in training value functions) than *Primal-RL*, however, in next sections we show several limitations that prevent leveraging their full potential.

## 2.2 LIMITATIONS OF CURRENT DUAL-RL METHODS

**Incorrect distribution ratio estimation** Notice that the nonlinear term in objective (2) contains the Bellman residual term $\mathcal{T}_r V(s,a) - V(s)$. In residual-gradient update, both $V(s')$ and $-V(s)$ would contribute to the gradient update, which causes slow convergence and a gradient conflict issue (Baird, 1995; Zhang et al., 2019b), especially in the offline setting where the gradient on $-V(s)$ is crucial for implicit maximization to find the best action (Mao et al., 2024a). To remedy this, one often uses a prevalent semi-gradient technique in RL that estimates $\mathcal{T}_r V(s,a)$ with $Q(s,a)$. The update of $Q(s,a)$ and $V(s)$ in semi-gradient *Dual-RL* methods are as follows (see Appendix A.1 for details).

$$\min_V \mathbb{E}_{(s,a) \sim d^{\mathcal{D}}}\left[V(s) + \alpha f_p^*\left([Q(s,a) - V(s)]/\alpha\right)\right] \tag{3}$$

$$\min_Q \mathbb{E}_{(s,a,s') \sim d^{\mathcal{D}}}\left[\left(r(s,a) + \gamma V(s') - Q(s,a)\right)^2\right]. \tag{4}$$

Note that $1-\gamma$ is omitted and the initial state distribution $d_0$ is replaced with dataset distribution $d^{\mathcal{D}}$ to stabilize learning (Kostrikov et al., 2020). This learning objective of $V$ can be viewed as an implicit maximizer to estimate $\sup_{a \sim \mu(a|s)} Q(s,a)$ (Xu et al., 2023; Sikchi et al., 2023b), the first term pushes down $V$-values while the second term pushes up $V$-values if $Q - V > 0$. In addition, because $f_p^*$ is non-linear and $\mathcal{D}$ usually cannot cover all possible $s'$, using the semi-gradient update also helps alleviate biased gradient estimation caused by single-sample estimation of $\mathcal{T}_r$ in objective (2). However, as we show later in Section 3, using semi-gradient update changes the fixed-point of the learning objective, preventing us from learning the true visitation distribution ratio.

**Data-regularized policy extraction** One intriguing property of *Dual-RL* is that, although the learning objective doesn't contain the policy, it actually learns an implicit optimal policy through the visitation distribution ratio between the optimized and behavior policy as $\pi^*(a|s) \propto \frac{d^*(s,a)}{d^{\mathcal{D}}(s,a)}\mu(a|s)$. One way to extract an explicit policy is minimizing the KL divergence (Boyd et al., 2004) between $\pi$ and $\pi^*$, which is equivalent to performing advantage-weighted behavior cloning (Peng et al., 2019) using the visitation distribution ratio as shown:

$$w^*(s,a) := \frac{d^*(s,a)}{d^{\mathcal{D}}(s,a)} = \max\left(0, (f')^{-1}\left((Q^*(s,a) - V^*(s))/\alpha\right)\right) \tag{5}$$

$$\pi = \arg\max_\pi \mathbb{E}_{(s,a) \sim d^{\mathcal{D}}}\left[w^*(s,a) \cdot \log\pi(a|s)\right]. \tag{6}$$

The explicit policy is parametrized as a unimodal Gaussian distribution in order to compute $\log\pi(a|s)$ (Haarnoja et al., 2018). However, this may cause some mode-covering behavior and deviation from the optimal mode if some suboptimal transitions are assigned non-zero weight (Ke et al., 2019). One way to solve this problem is using more expressive generative models like diffusion models (Ho et al., 2020; Song et al., 2020) to fit the multimodal behavior distribution and select the optimal mode using guided sampling (Mao et al., 2024b). However, the success depends on both accurate behavior modeling and correct sampling guidance, and also bring additional training/evaluation burden.

## 3 ITERATIVE DUAL-RL: TOWARDS OPTIMAL DISCRIMINATOR-WEIGHTED BEHAVIOR CLONING

In this section, we introduce our proposed approach, *Iterative Dual-RL* (IDRL). IDRL presents a principled solution to fix the aforementioned two limitations in *Dual-RL*. We start by deriving

the result that semi-gradient *Dual-RL* learns the action distribution ratio between the optimized and behavior policy; we then propose a correction to recover the true visitation distribution ratio based on the obtained action distribution ratio. Leveraging the learned visitation distribution ratios, we present an iterative self-distillation method to approach the optimal visitation distribution ratio. Theoretical analysis shows how using iterations help *Dual-RL* get a better imitation result by adopting curriculum-refined dataset filtering. Finally, we use a motivating toycase to validate the usefulness of IDRL in successfully extracting the optimal visitation distribution in the dataset.

### 3.1 TRUE VISITATION DISTRIBUTION ESTIMATION

First, we show that semi-gradient *Dual-RL* changes the fixed point of optimization and gives a solution that learns the action distribution ratio between the optimized and behavior policy rather than the expected visitation distribution ratio.

**Proposition 1.** *Semi-gradient Dual-RL only learns $w^*(a|s) = \frac{\pi^*(a|s)}{\mu(a|s)}$ instead of $w^*(s,a) = \frac{d^*(s,a)}{d^{\mathcal{D}}(s,a)}$.*

The result can be obtained by setting the derivative of objective (3) w.r.t $V(s)$ to zero, where we have $\forall s \sim d^{\mathcal{D}}, 1 + \mathbb{E}_{a \sim \mu}[-\max(0, (f')^{-1}((Q^*(s,a) - V^*(s))/\alpha))] = 0$, and leveraging the fact that $(f_p^*)'(x) = \max(0, (f')^{-1}(x))$.

There are two main shortcomings of learning an action-distribution ratio. (1) Action distribution ratio does not reason whether states are ever visited by the optimal policy (are 'good'), therefore it may assign positive weights to actions at bad states. Under the function approximation setting where states may share similar representations, this is more likely to cause incorrect generalization to suboptimal patterns generated by 'bad' states. This problem is especially severe when the dataset is dominated by a large portion of suboptimal data. (2) Iteratively filtering the dataset with action ratios can result in a fragmented dataset as some weights can be zero. For instance, if a transition is assigned a zero weight, the preceeding transition in trajectory is fragmented and does not have a backup target. Deep RL algorithms are not well suited to learn from such fragmented trajectories as they will overestimate the value of missing transitions.

We reframe as an off-policy evaluation (OPE) problem (Uehara et al., 2022) to recover the state-action visitation distribution ratio given the action distribution ratio. Inspired by techniques from Nachum & Dai (2020), we construct and solve the following constrained optimization problem

$$\max_{d \geq 0} -h(d) \quad \text{s.t. } d(s) = (1 - \gamma)d_0(s) + \gamma \sum_{\tilde{s},\tilde{a}} d(\tilde{s})\pi^*(\tilde{a}|\tilde{s})\mathcal{P}(s|\tilde{s},\tilde{a}), \forall s \in \mathcal{S}. \tag{7}$$

In this optimization problem, the $|\mathcal{S}|$ constraints uniquely determine $d = d^*$ while $h$ serves as a function that could make the optimization problem more approachable and stable when applying duality; common choices of $h$ include $h(d) := 0$ or $h(d) := D_f(d\|d^{\mathcal{D}})$.

The above objective (7) requires fitting an explicit policy $\pi^*$. We leverage the fact that $\pi^*(a|s)$ can be estimated via importance sampling $w^*(a|s) \cdot \mu(a|s)$ to propose a way to learn visitation ratios while only ever sampling from $\mu$ in Theorem 1. Theorem 1 presents a tractable OPE objective by applying Fenchel-Rockafellar duality (Rockafellar, 1970) to Eq.(7) under $h(d) := D_f(d\|d^{\mathcal{D}})$.

**Theorem 1.** *Given an action distribution ratio $w^*(a|s)$, we can recover its corresponding state visitation distribution ratio $w^*(s)$ as*

$$w^*(s) := \frac{d^*(s)}{d^{\mathcal{D}}(s)} = \max\left(0, (f')^{-1}\left(\mathbb{E}_{a \sim \mu}\left[w^*(a|s)(\mathcal{T}U^*(s,a) - U^*(s))\right]\right)\right), \tag{8}$$

*where $U^*$ is the optimal solution of the dual form of (7) as following,*

$$\min_{U} \mathbb{E}_{(s,a) \sim d^{\mathcal{D}}}\left[U(s) - \mathcal{T}U(s,a)\right] + \mathbb{E}_{s \sim d^{\mathcal{D}}}\left[f_p^*\left(\mathbb{E}_{a \sim \mu}\left[w^*(a|s)\left(\mathcal{T}U(s,a) - U(s)\right)\right]\right)\right]. \tag{9}$$

**Obtaining an unbiased estimator:** In objective (9), the first term is easy to estimate, however, an unbiased estimation of the second term along with $w^*(s)$ is generally non-trivial due to the expectation inside a non-linear function $f_p^*$ or $(f')^{-1}$. We show that Lemma 1 can be used to obtain an unbiased estimate for the second term above.

**Lemma 1.** *Given a random variable $X$ and its corresponding distribution $P(X)$, for any convex function $f(x)$, the following problem is convex and the optimal solution is $y^* = (f')^{-1}(\mathbb{E}_{x \sim P(X)}[g(x)])$.*

$$\min_y \mathbb{E}_{x \sim P(X)}[f(y) - g(x) \cdot y].$$

The proof follows by setting the derivative of the objective w.r.t $y$ to zero. Substituting $P(x)$ as $\mu$ and $g(x)$ as $w^*(a|s)(\mathcal{T}U(s,a) - U(s))$, we can get the following result:

**Theorem 2.** *The unbiased estimate of $(f')^{-1}\big(\mathbb{E}_{a \sim \mu}\left[w^*(a|s)(\mathcal{T}U(s,a) - U(s))\right]\big)$ is the optimal solution $W^*(s)$ of the following optimization problem:*

$$\min_W \mathbb{E}_{(s,a) \sim d^{\mathcal{D}}}\Big[f(W(s)) - w^*(a|s)(\mathcal{T}U(s,a) - U(s)) \cdot W(s)\Big].$$

This learning objective provides an unbiased estimator by only using $(s,a,s')$ samples from the dataset, as two expectations that cause biased gradient estimation ($\mathbb{E}_{a \sim \mu}$ and $\mathbb{E}_{s'}$ in $\mathcal{T}U$) are all linear. Note that this objective is convex with respect to $W$, which indicates a guarantee of convergence to $W^*$ under mild assumptions.

Using the fact that $(f_p^*)'(x) = \max(0, (f')^{-1}(x))$, we are ready to present a surrogate objective for the second term in Eq.(9) that admits the same gradient w.r.t $U$, by using the following equation.

$$\mathbb{E}_{s \sim d^{\mathcal{D}}}\Bigg[(f_p^*)'\Big(\mathbb{E}_{a \sim \mu}\Big[w^*(a|s)\left(\mathcal{T}U(s,a) - U(s)\right)\Big]\Big) \cdot \mathbb{E}_{a \sim \mu}\Big[w^*(a|s)\nabla_U\left(\mathcal{T}U(s,a) - U(s)\right)\Big]\Bigg]$$

$$= \nabla_U \mathbb{E}_{(s,a) \sim d^{\mathcal{D}}}\Big[\max(0, W^*(s)) \cdot w^*(a|s)\left(\mathcal{T}U(s,a) - U(s)\right)\Big].$$

Finally with the reductions shown above, learning Eq.(9) in an unbiased way using sample-based estimation amounts to solving the following two optimization problems jointly:

$$\min_W \mathbb{E}_{(s,a) \sim d^{\mathcal{D}}}\Big[f(W(s)) - w^*(a|s)(\mathcal{T}U(s,a) - U(s)) \cdot W(s)\Big], \tag{10}$$

$$\min_U \mathbb{E}_{(s,a) \sim d^{\mathcal{D}}}\Big[U(s) - \mathcal{T}U(s,a)\Big] + \mathbb{E}_{(s,a) \sim d^{\mathcal{D}}}\Big[\max(0, W(s)) \cdot w^*(a|s)(\mathcal{T}U(s,a) - U(s))\Big]. \tag{11}$$

In (11), the learning objective introduces sparsity in learning the value function (Xu et al., 2023). The first term pushes down value residuals while the second term pushes up value residuals if $w^*(s,a) > 0$. This makes the residuals of $(s,a)$ under the regularized optimal visitation distribuion to be high. Note that the learning of $U$ and $W$ introduces no extra hyperparameters that are prevalent in offline RL. After getting the optimal $U^*$ and $W^*$, we can obtain the corrected state-action visitation distribution ratio as:

$$\frac{d^*(s,a)}{d^{\mathcal{D}}(s,a)} = w^*(s,a) = w^*(s) * w^*(a|s) = \max\big(0, W^*(s)(f')^{-1}\big(Q^*(s,a) - V^*(s)\big)\big). \tag{12}$$

### 3.2   LEARN THE OPTIMAL DISCRIMINATOR WEIGHT BY ITERATIVE SELF-DISTILLATION

Although we get the correct visitation distribution ratio in *Dual-RL*, that ratio is not the true optimal visitation distribution ratio (i.e. $d^* \neq d^E$) as it is regularized by the offline data distribution. We thus propose an iterative self-distillation way to break the regularization barrier, our key insight is that we can leverage the learned (regularized) optimal visitation distribution ratio after one iteration of *Dual-RL* to refine the offline dataset to a new one, which will be used for another iteration of *Dual-RL*. This procedure can be repeated several times to approach the true optimal visitation distribution ratio.

More specifically, we extend *Dual-RL* to $M$ iterations ($M \geq 1$) and use $\chi^2$-divergence as $D_f$. After the $i$-th iteration of *Dual-RL*, we select a support of dataset $\mathcal{D}_i$ that has non-zero probability mass in optimal regularized visitation ($w_i^*(s,a) > 0$), and run the $i+1$-th iteration of *Dual-RL*. The visitation distribution ratio at the last iteration $w_M^*(s,a)$ is used for learning the policy by weighted behavior cloning. We term this iterative method as *iterative Dual-RL* (IDRL), the pseudo-code is presented in Algorithm 1. IDRL uses multiple iterations of *Dual-RL* to break the regularization barrier that impedes offline RL to learn performant policies by selecting a support of offline dataset without suffering overestimation. Note that breaking the regularization barrier is hard with offline *Primal-RL* algorithms as they do not reason about visitation distributions, thus making it impossible to refine the offline dataset used for regularization.

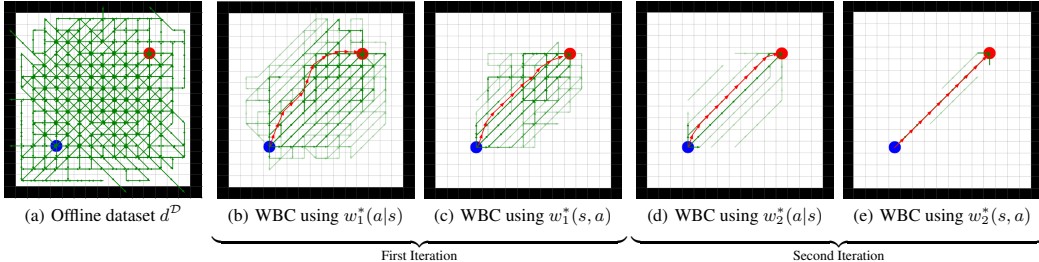

(a) Offline dataset $d^{\mathcal{D}}$    (b) WBC using $w_1^*(a|s)$    (c) WBC using $w_1^*(s,a)$    (d) WBC using $w_2^*(a|s)$    (e) WBC using $w_2^*(s,a)$

First Iteration        Second Iteration

Figure 2: IDRL in a grid-world domain. The initial state (blue) and the goal state (red) define the task, with the green arrows representing remaining transitions from the dataset. The opacity of the green arrows denotes magnitude of the weights from the respective distribution ratios. Red arrows depict the trajectories generated by policies obtained through weighted-BC with the estimated distribution ratios. (a) Original dataset. (b) and (d) show the results after filtering the dataset based on the learned policy ratio (Uncorrected *Dual-RL* visitation ratio) $w(a|s)$ in the first and second iterations, respectively. (c) and (e) demonstrate the subsequent filtering using the state-action visitation distribution ratio $w(s,a)$, which is computed by combining $w(a|s)$ and $w(s)$ (IDRL correction). This process reveals that the method progressively focuses on the most relevant transitions, enabling the recovery of a near-optimal visitation distribution ratio after 2 iterations.

The filtering procedure in IDRL can be approximately viewed as replacing each iteration's behavior visitation distribution to a better one (i.e., the regularized optimal visitation distribution at this iteration): $D_f(d\|d_k^*) \Rightarrow D_f(d\|d_{k+1}^*)$. Intuitively, this could boost performance improvement, especially on suboptimal datasets. Formally, we give a theoretical analysis of *IDRL* based on previous analysis of behavior cloning, we aim to 1) give the behavior cloning performance bound of IDRL, and 2) analyze how iterations will influence this bound.

**Theorem 3.** *Given horizon length $H$ and the dataset sample size of $\mathcal{D}$ as $N_{\mathcal{D}}$, the behavior cloning performance bound of IDRL at iteration $k$ is given by*

---

**Algorithm 1** Iterative Dual-RL (IDRL)

1: Initialize value functions $Q_{\phi_1}, V_{\phi_2}, U_{\psi_1}, W_{\psi_2}$, policy network $\pi_\theta$, require $\alpha$ and dataset $\mathcal{D}_1 = \mathcal{D}$
2: **for** $k = 1, 2, \cdots, M$ **do**
3:      **for** $t = 1, 2, \cdots, N_1$ **do**
4:          Sample transitions $(s, a, r, s') \sim \mathcal{D}_k$
5:          Update $Q_{\phi_1}$ and $V_{\phi_2}$ by (4) and (3)
6:      **end for**
7:      Get action ratio $w_k(a|s)$ by Eq.(5)
8:      **for** $t = 1, 2, \cdots, N_2$ **do**
9:          Sample transitions $(s, a, s') \sim \mathcal{D}_k$
10:          Update $U_{\psi_1}$ and $W_{\psi_2}$ by (11) and (12)
11:      **end for**
12:      Get state-action ratio $w_k(s,a)$ by Eq.(12) and $\mathcal{D}_{k+1} = \{(s,a,r,s') \in \mathcal{D}_k \mid w_k(s,a) > 0\}$
13: **end for**
14: Learn $\pi_\theta$ by Eq.(6) using $\mathcal{D}_M$ and $w_M(s,a)$

---

$$V(\pi) = V(\mathcal{D}_{k+1}) - \mathcal{O}\left(\frac{|\mathcal{S}|H^2}{N_{\mathcal{D}_{k+1}} + N_{\mathcal{D}_k - \mathcal{D}_{k+1}}/\max_s w_{k+1}^*(s)}\right).$$

**Theorem 4.** *We have $V(\mathcal{D}_{k+1}) \geq V(\mathcal{D}_k)$ after the $k$-th iteration of IDRL.*

Theorem 3 is derived by framing weighted-BC as selecting expert data $\mathcal{D}_{k+1}$ from suboptimal data $\mathcal{D}_k$ using discriminator-weighted behavior cloning (Xu et al., 2022b; Li et al., 2024). From Theorem 3 we can see that if $\mathcal{D}_{k+1}$ is closer to the expert distribution, $V(\mathcal{D}_{k+1})$ increases, however, $N_{\mathcal{D}_{k+1}}$ decreases, so the second term may also increase. This hints a trade-off by adjusting $\mathcal{D}_{k+1}$ to maximize the behavior cloning performance bound. Theorem 4 shows that IDRL uses iterations to efficiently find the optimal choice of $\mathcal{D}_k$ by ensuring a monotonic improvement over $V(\mathcal{D}_k)$. This result is important as the above property is missing from classical offline RL; decreasing $\alpha$ may be expected to produce similar empirical results like Theorem 3; however, decreasing $\alpha$ does not have a monotonic improvement guarantee like Theorem 4, and suffers from overestimation in practice.

**IDRL in a gridworld toycase** We demonstrate the effectiveness of IDRL through a simple grid-world experiment. The environment consists of continuous state and action space. However, we collect offline data in a discrete manner. We use a discrete behavior policy that takes action randomly from $(\uparrow, \downarrow, \leftarrow, \rightarrow)$ to collect 500 transitions. Note that the weighted-BC policy is learned to be continuous with a unimodal gaussian distribution. We use this toycase to mimic the scenario in offline continuous control problem under function approximation, where similar states share the same representation in latent space so one latent state may have multiple behavior actions. In this case

Table 1: Averaged normalized scores of IDRL against other baselines. We report the average normalized scores at the end of training with standard deviation across 7 random seeds. We highlight the top score (integer-level). IDRL matches or outperforms previous SOTA *Primal-RL* and *Dual-RL* methods on almost all tasks.

| D4RL Dataset | Primal (Max-Q) | | | | Dual (Weighted-BC) | | | | |
|---|---|---|---|---|---|---|---|---|---|
| | TD3+BC | CQL | ReBRAC | Diffusion-QL | X%-BC | IQL | SQL | O-DICE | IDRL (ours) |
| halfcheetah-m | 48.3 | 44.0 | 64.0 ±1.0 | 51.1 ±0.5 | 42.5 | 47.4 | 48.3 ±0.2 | 47.4 ±0.2 | 58.4 ±0.1 |
| hopper-m | 59.3 | 58.5 | 102.2 ±1.0 | 90.5 ±4.6 | 56.9 | 66.3 | 75.7 ±3.4 | 86.1 ±4.0 | 99.5 ±0.4 |
| walker2d-m | 83.7 | 72.5 | 82.5 ±3.6 | 87.0 ±0.9 | 75.0 | 72.5 | 84.2 ±4.6 | 84.9 ±2.3 | 92.8 ±0.3 |
| halfcheetah-m-r | 44.6 | 45.5 | 51.0 ±0.8 | 47.8 ±0.3 | 40.6 | 44.2 | 44.8 ±0.7 | 44.0 ±0.3 | 58.0 ±0.3 |
| hopper-m-r | 60.9 | 95.0 | 98.1 ±5.3 | 101.3 ±0.6 | 75.9 | 95.2 | 99.7 ±3.3 | 99.9 ±2.7 | 101.7 ±0.7 |
| walker2d-m-r | 81.8 | 77.2 | 77.3 ±7.9 | 95.5 ±1.5 | 62.5 | 76.1 | 81.2±3.8 | 83.6±4.1 | 84.4 ±0.4 |
| halfcheetah-m-e | 90.7 | 90.7 | 101.1 ±5.2 | 96.8 ±0.3 | 92.9 | 86.7 | 94.0 ±0.4 | 93.2 ±0.6 | 98.1 ±0.2 |
| hopper-m-e | 98.0 | 105.4 | 107.0 ±6.4 | 111.1 ±1.3 | 110.9 | 101.5 | 110.8 ±2.2 | 110.8 ±0.6 | 111.0 ±0.4 |
| walker2d-m-e | 110.1 | 109.6 | 111.6 ±0.3 | 110.1 ±0.3 | 109.0 | 110.6 | 110.0 ±0.8 | 110.0 ±0.2 | 113.9 ±0.1 |
| antmaze-u | 78.6 | 84.8 | 97.8 ±1.0 | 85.5 ±1.9 | 62.8 | 85.5 | 92.2 ±1.4 | 94.1 ±1.6 | 99.5 ±0.3 |
| antmaze-u-d | 71.4 | 43.4 | 88.3 ±13.0 | 66.7 ±4.0 | 50.2 | 66.7 | 74.0 ±2.3 | 79.5 ±3.3 | 88.5 ±1.2 |
| antmaze-m-p | 10.6 | 65.2 | 84.0 ±4.2 | 72.2 ±5.3 | 5.4 | 72.2 | 80.2 ±3.7 | 86.0 ±1.6 | 93.0 ±1.1 |
| antmaze-m-d | 3.0 | 54.0 | 76.3 ±13.5 | 71.0 ±3.2 | 9.8 | 71.0 | 79.1 ±4.2 | 82.7 ±4.9 | 86.5 ±3.9 |
| antmaze-l-p | 0.2 | 38.4 | 60.4 ±26.1 | 39.6 ±4.5 | 0.0 | 39.6 | 53.2 ±4.8 | 55.9 ±3.9 | 60.3 ±3.4 |
| antmaze-l-d | 0.0 | 31.6 | 54.4 ±25.1 | 47.5 ±4.4 | 6.0 | 47.5 | 52.3 ±5.2 | 54.6 ±4.8 | 54.2 ±3.8 |
| kitchen-c | - | 43.8 | 77.2 ±8.3 | 84.0 ±7.4 | 33.8 | 61.4 | 76.4 ±8.7 | 75.0 ±6.6 | 80.5 ±3.8 |
| kitchen-p | - | 49.8 | 68.5 ±10.6 | 60.5 ±6.9 | 33.9 | 46.1 | 72.5 ±7.4 | 72.8 ±4.3 | 74.3 ±4.4 |
| kitchen-m | - | 51.0 | 55.3 ±9.2 | 62.6 ±5.1 | 47.5 | 52.8 | 67.4 ±5.4 | 65.8 ±3.6 | 67.8 ±2.1 |

assigning positive weights to suboptimal transitions will deteriorate the weighted-BC result. Figure 2 walks through how applying IDRL subsequently filters the offline dataset towards optimal visitation thus learning an optimal policy, while using only action ratios lead to suboptimal performance.

## 4  EXPERIMENTS

In this section, we present empirical evaluations of IDRL on different kinds of offline datasets. We first evaluate IDRL on D4RL benchmark offline RL datasets (Fu et al., 2020) and compare against several SOTA *Primal-RL* and *Dual-RL* baseline algorithms. However, D4RL datasets are collected from policies that enable methods with strong distribution constraints to already have strong performance. As a consequence, we also find that IDRL only needs one or two iterations to achieve best performance. To further show the benefits of IDRL, we test IDRL on more realistic datasets where more iterations are needed to find the optimal visitation distribution. We choose corrupted demonstrations where few expert demonstrations are mixed with a large portion of random data (Xu et al., 2022b; Hong et al., 2023b). Experimental details are shown in Appendix C.

### 4.1  RESULTS ON OFFLINE RL

We first evaluate IDRL on the D4RL benchmark (Fu et al., 2020) and compare it with several related algorithms. For the evaluation tasks, we select Mujoco locomotion tasks, Antmaze navigation tasks and Kitchen tasks which require both locomotion and navigation. While Mujoco tasks are popular in offline RL, Antmaze and Kitchen tasks are more challenging due to their stronger need for selecting optimal parts of different trajectories to perform stitching. For baseline algorithms, we selected state-of-the-art methods not only from *Primal-RL* methods (that select best policy modes given by Max-$Q$), but *Dual-RL* methods (weighted-BC). *Primal-RL* baselines includes TD3+BC (Fujimoto & Gu, 2021), CQL (Kumar et al., 2020), ReBRAC (Tarasov et al., 2024) and Diffusion-QL (Wang et al., 2023) which has strong policy expressivity by using diffusion models. *Dual-RL* baselines include IQL (Kostrikov et al., 2021b), SQL (Xu et al., 2023) and O-DICE (Mao et al., 2024a). The results of IQL and SQL reflect the performance of using semi-gradient update as they apply action-level behavior constraints. Notably, O-DICE is a recently proposed algorithm that stands out among various *Dual-RL* methods. Although it claims to achieve the correct state-action visitation distribution ratio, it introduces another hyperparameter $\eta$ which is hard to tune in practice. We also include the results of using X%-BC, which is a filtered version of BC that runs behavior cloning on only the top X% high-return trajectories in the dataset. This comparision is to show the necessity of behavior

cloning based on transition-wise selection rather than trajectory-wise. We select the best results by spanning X in $\{2, 5, 10\}$.

The results are shown in Table 1, it can be seen that IDRL matches or outperforms all previous SOTA algorithms on almost all D4RL tasks. This suggests that IDRL can effectively learn a strong policy from the dataset, even in challenging Antmaze and Kitchen tasks that require RL to "stitch" good trajectories. The large performance boost between IDRL and previous *Dual-RL* methods, especially on sub-optimal datasets that contain less or few near-optimal trajectories, demonstrates the benefits of iteratively finding the optimal visitation distribution ratio. The consistently better performance compared with IDRL and *Primal-RL* baselines demonstrates the essentiality of in-sample value learning, which causes fewer overestimation errors. All these results reveal the power of using an optimal discriminator-weighted imitation view of solving offline RL.

**Ablation study** To show both components are necessary in IDRL, we perform an ablation study where we compare the performance of IDRL with (IDRL w/ $M = 1$), and the ablation that uses $w^*(a|s)$ to filter datasets (IDRL w/ $w^*(a|s)$). It can be shown in Table 2 that using more

Table 2: Ablation study of IDRL.

| Ablation | Mujoco | Antmaze | Kitchen |
|---|---|---|---|
| IDRL | 90.8 | 80.3 | 74.8 |
| IDRL w/ $M = 1$ | 73.2 | 68.8 | 70.5 |
| IDRL w/ $w^*(a|s)$ | 56.8 | 51.3 | 67.8 |

iterations boosts the performance by finding the best support of the dataset to do imitation learning, as discussed in Theorem 3. This also validates the importance of using the correct state-action visitation distribution ratio as using action distribution ratio will get significantly worse results due to overestimation errors caused by fragmented trajectories. This ablation study justifies the benefits of both improvements to *Dual-RL*.

## 4.2 RESULTS ON CORRUPTED DEMONSTRATIONS

In this experiment, we aim to demonstrate the effectiveness of IDRL in recovering the true visitation distribution from datasets that contain a significant amount of noisy or suboptimal transitions. The experiment involves creating a "mixed" dataset by combining random and expert policy generated transitions from Mujoco datasets in D4RL at varying expert ratios, thereby simulating a dataset collected with low-performing behavior policies. The purpose of this experiment is to tackle a common challenge

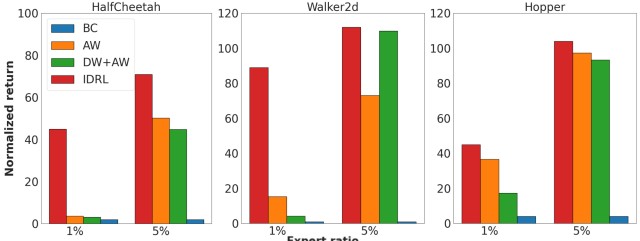

Figure 3: Performance of naive BC, Advantage-Weighted (AW), Density-Weighted initialized with AW (DW+AW) and IDRL on mixed datasets created by combining varying percentages of expert transitions with random transitions from D4RL Mujoco datasets. For IDRL, this table shows the results using three iterations.

in offline RL algorithms: their tendency to anchor the learned policy to the dataset's behavior policy. As shown by previous work (Hong et al., 2023a;b), while this anchoring works well when the behavior policy is high-performing, it becomes problematic in datasets dominated by low-performing trajectories. In such heteroskedastic and corrupted datasets, where only a few trajectories are high-performing, most offline RL algorithms struggle to recover the optimal policy because they heavily rely on the behavior policy present in the data. By applying IDRL iteratively to these mixed datasets, we aim to remove suboptimal transitions, gradually refining the dataset to better represent the optimal policy. Compare with the weighted sampling strategy proposed by previous paper (Hong et al., 2023a;b), iterative application of IDRL can exploit and combine the rare, high-performing transitions while filtering out suboptimal transitions, effectively learning a near-true visitation distribution ratio that enables the recovery of an optimal policy, even in challenging, noisy offline settings.

## 5 RELATED WORK

**Primal-RL and Dual-RL approaches:** Offline RL requires trading off between unconstrained policy improvement and staying 'close' to the offline data distribution, as choosing actions arbitrarily can lead to requiring evaluation of policies that go beyond the support of offline dataset and have incorrect evaluation especially with overestimation bias (Fujimoto et al., 2018) present in Bellman

optimality backups. To tackle this distributional shift problem, most classical/primal offline RL methods augment existing off-policy RL methods with an action-level behavior regularization term that prevents the learned policy from deviating too far from the dataset policy. Action-level regularization can appear explicitly as divergence penalties (Wu et al., 2019; Kumar et al., 2019; Xu et al., 2021; Fujimoto & Gu, 2021; Cheng et al., 2023; Li et al., 2023), implicitly through weighted behavior cloning (Wang et al., 2020; Nair et al., 2020), or more directly through careful parameterization of the policy (Fujimoto et al., 2019; Zhou et al., 2020). Another way to apply action-level regularization is by learning pessimistic value functions that encourage staying near the behavioral distribution and being pessimistic about OOD actions (Kumar et al., 2020; Kostrikov et al., 2021a; Xu et al., 2022c; 2023; Wang et al., 2023; Niu et al., 2022). Several works also incorporate action-level regularization through the use of uncertainty (An et al., 2021; Bai et al., 2021) or distance function (Li et al., 2022a). Another line of methods, on the contrary, imposes action-level regularization by performing a form of imitation learning on the dataset.When the dataset is high-quality (contains high return trajectories), we can simply clone or filter dataset actions to extract useful transitions (Xu et al., 2022b; Chen et al., 2020; Zhang et al., 2023; Zheng et al., 2024), or directly filter individual transitions based on how advantageous they could be under the behavior policy and then clone them (Brandfonbrener et al., 2021; Xu et al., 2022a). Prior attempts to include state-action level behavior regularization (Li et al., 2022b; Zhang et al., 2022) have required computationally costly extra steps of model-based OOD state detection (Li et al., 2022b; Zhang et al., 2022). Most offline RL methods rely on a unimodal Gaussian policy that suffer from mode-covering behavior during policy optimization potentially outputting suboptimal policies. The use of expressive generative models allows us to move past this limitation (Hansen-Estruch et al., 2023; Mao et al., 2024b) by their ability to match value functions modes accurately, albeit at a high computational cost and a slow inference speed.

Unlike the above Primal-RL approaches, Dual RL approaches reason in the visitation space for maximizing returns and directly regularize the state-action visitation divergence with the offline dataset. As a by-product, these methods also promise to learn the ratio of optimal regularized visitation distribution with offline data visitation. Numerous works have shown their utility in off-policy evaluation (Nachum et al., 2019a; Zhang et al., 2019a; 2020), offline policy selection (Yang et al., 2020), off-policy RL (Nachum et al., 2019b; Lee et al., 2021; Sikchi et al., 2023b), safe RL (Lee et al., 2022), GCRL Ma et al. (2022b); Sikchi et al. (2023a), and imitation learning (Kostrikov et al., 2020; Zhu et al., 2020; Garg et al., 2021; Kim et al., 2021; Ma et al., 2022a; Sikchi et al., 2024).

**Connection between offline RL and imitation learning:** Offline RL and imitation learning share deep connections. Rashidinejad et al. (2021) studies the connection theoretically from the lens of data composition to analyze optimality rates; Sikchi et al. (2023b) shows that imitation learning becomes a special case of RL with reward set to zero in the dual RL framework. Other works (Kumar et al., 2022) have empirically evaluated when imitation learning in the form of behavior cloning is preferred over offline RL. Our work, motivated by the simple experiment that having access to expert data is sufficient to outperform current offline RL methods, proposes a smooth and principled interpolation that refines offline datasets in offline RL to be closer to expert state-action visitation distribution. This is opposed to prior works that iteratively relax policy regularization (Li et al., 2023; Hu et al., 2023; Liu et al., 2024) which only consider actions, potentially learning suboptimal actions at states never visited by the expert and lacking a principled reduction to expert visitation distribution.

## 6 CONCLUSION AND LIMITATIONS

In this paper, we provide an optimal discriminator-weighted imitation view of solving offline RL. Motivated by a simple experiment that finds the effectiveness of optimal discriminator-weighted behavior cloning, we build on the result of *Dual-RL* and propose *iterative Dual-RL* (IDRL) that aims to fix two pitfalls of current *Dual-RL* methods. IDRL iteratively filters out suboptimal transitions and extracts a policy with weighted behavior cloning on that subdataset. We give both theoretical and empirical justification for our approach. IDRL achieves SOTA results on various kinds of datasets, including D4RL benchmark datasets and noisy demonstrations where all other offline RL methods fail. One limitation of IDRL is the increase of training time due to running multiple iterations. Another limitation is that IDRL may suffer from generalization issues when the data size is small. One future work is to extend IDRL to the online setting where *Dual-RL* naturally serves as a principled off-policy method.

ACKNOWLEDGEMENT

This work is supported by NSF 2340651, NSF 2402650, DARPA HR00112490431, and ARO W911NF-24-1-0193. The authors would like to thank the ICLR reviewers for their constructive feedback.

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

## A  AN EXTENDED INTRODUCTION OF DUAL-RL METHODS

### A.1  DERIVATION OF DUAL-RL

Dual-RL algorithms consider the following regularized RL problem as a convex programming problems with Bellman-flow constraints and apply Fenchel-Rockfeller duality or Lagrangian duality to solve it. The regularization term aims at imposing visitation distribution constraints (Nachum & Dai, 2020; Lee et al., 2021; Mao et al., 2024a).

$$\max_{\pi} \mathbb{E}_{(s,a)\sim d^{\pi}}[r(s,a)] - \alpha D_f(d^{\pi}(s,a)\|d^{\mathcal{D}}(s,a)).$$

$D_f(d^{\pi}(s,a)\|d^{\mathcal{D}}(s,a))$ is the $f$-divergence which is defined with $D_f(P\|Q) = \mathbb{E}_{\omega\in Q}\left[f\left(\frac{P(\omega)}{Q(\omega)}\right)\right]$.
Directly solving $\pi^*$ is impossible because it's intractable to calculate $d^{\pi}(s,a)$. However, one can change the optimization variable from $\pi$ to $d^{\pi}$ because of the bijection existing between them. Then with the assistance of Bellman-flow constraints, we can obtain an optimization problem with respect to $d$:

$$\max_{d\geq 0} \mathbb{E}_{(s,a)\sim d}[r(s,a)] - \alpha D_f(d(s,a)\|d^{\mathcal{D}}(s,a))$$

$$\text{s.t.} \sum_{a\in\mathcal{A}} d(s,a) = (1-\gamma)d_0(s) + \gamma \sum_{(s',a')} d(s',a')p(s|s',a'), \forall s\in\mathcal{S}.$$

Note that the feasible region has to be $\{d : \forall s \in \mathcal{S}, a \in \mathcal{A}, d(s,a) \geq 0\}$ because $d$ should be non-negative to ensure a valid corresponding policy. After applying Lagrangian duality, we can get the following optimization target following (Lee et al., 2021):

$$\min_{V(s)} \max_{d\geq 0} \mathbb{E}_{(s,a)\sim d}[r(s,a)] - \alpha D_f(d(s,a)\|d^{\mathcal{D}}(s,a))$$

$$+ \sum_{s} V(s)\Big((1-\gamma)d_0(s) + \gamma \sum_{(s',a')} d(s',a')p(s|s',a') - \sum_{a} d(s,a)\Big)$$

$$= \min_{V(s)} \max_{\omega\geq 0}(1-\gamma)\mathbb{E}_{d_0(s)}[V(s)]$$

$$+ \mathbb{E}_{s,a\sim d^{\mathcal{D}}}\Big[\omega(s,a)\Big(r(s,a) + \gamma \sum_{s'} p(s'|s,a)V(s') - V(s)\Big)\Big] - \alpha\mathbb{E}_{s,a\sim d^{\mathcal{D}}}\Big[f(\omega(s,a))\Big].$$

Here we denote $\omega(s,a)$ as $\frac{d(s,a)}{d^{\mathcal{D}}(s,a)}$ for simplicity. By incorporating the non-negative constraint of $d$ and again solving the constraint problem with Lagrangian duality, we can derive the optimal solution $w^*(s,a)$ for the inner problem and thus reduce the bilevel optimization problem to the following optimization problem:

$$\min_{V(s)}(1-\gamma)\mathbb{E}_{d_0(s)}[V(s)] + \alpha\mathbb{E}_{s,a\sim d^{\mathcal{D}}}\Big[f_p^*\big(\frac{r(s,a) + \gamma\sum_{s'}p(s'|s,a)V(s') - V(s)}{\alpha}\big)\Big].$$

Here $f_p^*$ is a variant of $f$'s convex conjugate as $f_p^* = \max\big(0, f'^{-1}(x)\big)(x) - f\big(\max\big(0, f'^{-1}(x)\big)\big)$.

### A.2  DIFFERENT DUAL-RL METHODS

Note that in the offline RL setting, the inner summation $\sum_{s'} p(s'|s,a)V(s')$ is usually intractable because of limited data samples. To handle this issue and increase training stability, semi-gradient Dual-RL methods usually use additional network $Q(s,a)$ to fit $r(s,a) + \gamma\sum_{s'} p(s'|s,a)V(s')$, by optimizing the following MSE objective:

$$\min_{Q} \mathbb{E}_{(s,a,s')\sim d^{\mathcal{D}}}\big[\big(r(s,a) + \gamma V(s') - Q(s,a)\big)^2\big].$$

In doing so, the optimization objective in A.1 can be replaced with:

$$\min_{V} \mathbb{E}_{s\sim d^0}[(1-\gamma)V(s)] + \mathbb{E}_{(s,a)\sim d^{\mathcal{D}}}\big[\alpha f^*\big([Q(s,a) - V(s)]/\alpha\big)\big]. \tag{13}$$

Also note that to increase the diversity of samples, one often extends the distribution of initial state $d_0$ to $d^{\mathcal{D}}$ by treating every state in a trajectory as initial state (Kostrikov et al., 2020).

Recently, another Dual-RL method O-DICE (Mao et al., 2024a) was proposed to improve semi-gradient Dual-RL. O-DICE uses orthogonal-gradient update to Dual-RL, which adds a projected backward gradient to semi-gradient. The projected backward gradient can be written as

$$\nabla_\theta^\perp V(s') = \nabla_\theta V(s') - \frac{\nabla_\theta V(s)^\top \nabla_\theta V(s')}{\|\nabla_\theta V(s)\|^2}\nabla_\theta V(s).$$

Intuitively, the projected backward gradient will not interfere with the forward (semi) gradient while still retaining information from the backward gradient. After getting the projected backward gradient, O-DICE adds it to the forward gradient term, resulting in a new gradient flow as

$$\nabla_\theta f^*\left(\hat{\mathcal{T}}V(s,a) - V(s))\right) := (f^*)'(r + \gamma V(s') - V(s))\left(\gamma \cdot \eta\nabla_\theta^\perp V(s') - \nabla_\theta V(s)\right),$$

where O-DICE uses one more hyperparameter $\eta > 0$ to control the strength of the projected backward gradient against the forward gradient. In theory, $\eta$ needs to be large enough to guarantee a convergence.

## B PROOFS

**Lemma 2.** $(1 - \gamma)\mathbb{E}_{s\sim d_0}[V(s)] = \mathbb{E}_{(s,a)\sim d^{\mathcal{D}}}[V(s) - \mathcal{T}V(s,a)].$

*Proof.* The proof can be simply obtained by using the definition of state visitation distribution $d_t$ at timestep $t$ following the behavior policy $\mu$.

$$(1 - \gamma)\mathbb{E}_{s\sim d_0}[V(s)] = (1 - \gamma)\sum_{t=0}^\infty \gamma^t\mathbb{E}_{s\sim d_t}[V(s)] - (1 - \gamma)\sum_{t=0}^\infty \gamma^{t+1}\mathbb{E}_{s\sim d_{t+1}}[V(s)]$$

$$= (1 - \gamma)\sum_{t=0}^\infty \gamma^t\mathbb{E}_{s\sim d_t}\left[V(s) - \gamma\mathbb{E}_{s'\sim T(s,a)}[V(s')]\right]$$

$$= \mathbb{E}_{(s,a)\sim d^{\mathcal{D}}}[V(s) - \mathcal{T}V(s,a)]$$

$\square$

**Theorem 1.** Given an action distribution ratio $w^*(a|s)$, we can recover its corresponding state visitation distribution ratio $w^*(s)$ as

$$w^*(s) := \frac{d^*(s)}{d^{\mathcal{D}}(s)} = \max\left(0, (f')^{-1}\left(\mathbb{E}_{a\sim\mu}[w^*(a|s)(\mathcal{T}U^*(s,a) - U^*(s))]\right)\right),$$

where $U^*$ is the optimal solution of the dual form of (7) as following,

$$\min_U \mathbb{E}_{(s,a)\sim d^{\mathcal{D}}}\left[U(s) - \mathcal{T}U(s,a)\right] + \mathbb{E}_{s\sim d^{\mathcal{D}}}\left[f_p^*\left(\mathbb{E}_{a\sim\mu}\left[w^*(a|s)\left(\mathcal{T}U(s,a) - U(s)\right)\right]\right)\right].$$

*Proof.* We first reframe (7) as

$$\max_d -g(-Ad) - h(d)$$

where $g(-Ad)$ corresponds to the linear constraints with respect to the adjoint Bellman operator,

$$g := \delta_{\{(1-\gamma)d_0\}} \quad \text{and} \quad A := \gamma \cdot \mathcal{T}_* - I.$$

When applying Fenchel-Rockafellar duality, the linear operator $A$ is transformed to its adjoint $A_* = \gamma \cdot \mathcal{T} - I$ and is used as an argument to the Fenchel conjugate $h_*(\cdot) = \mathbb{E}_{d^{\mathcal{D}}}[f_p^*(\cdot)]$ of $h$. At the same time, $g$ is replaced by its Fenchel conjugate $g_*(\cdot) = (1 - \gamma)\mathbb{E}_{d_0}[\cdot]$.

The dual problem is therefore given by

$$\min_U g_*(U) + h_*(A_*U)$$

$$= \min_U (1 - \gamma)\mathbb{E}_{s\sim d_0}[U(s)] + \mathbb{E}_{s\sim d^{\mathcal{D}}}\left[f_p^*\left(\mathbb{E}_{a\sim\mu}[w^*(a|s)\left(\mathcal{T}U(s,a) - U(s)\right)]\right)\right]$$

We can get Theorem 1 by replacing $(1 - \gamma)\mathbb{E}_{s\sim d_0}[U(s)]$ with $\mathbb{E}_{(s,a)\sim d^{\mathcal{D}}}[U(s) - \mathcal{T}U(s,a)]$ using Lemma 2.

$\square$

**Lemma 1.** Given a random variable $X$ and its corresponding distribution $P(X)$, for any convex function $f(x)$, the following problem is convex and the optimal solution is $y^* = (f')^{-1}(\mathbb{E}_{x \sim P(X)}[g(x)])$.

$$\min_y \mathbb{E}_{x \sim P(X)}[f(y) - g(x) \cdot y].$$

*Proof.* Taking the derivative of $L(y)$ with respect to $y$ gives:

$$\frac{d}{dy} L(y) = \mathbb{E}_{x \sim P(X)}[f'(y) - g(x)]$$

Setting the derivative to zero for optimality:

$$f'(y^*) = \mathbb{E}_{x \sim P(X)}[g(x)]$$

Since $f$ is convex, $f'$ is invertible, and the optimal $y^*$ is:

$$y^* = (f')^{-1}\left(\mathbb{E}_{x \sim P(X)}[g(x)]\right)$$

$\square$

**Theorem 3.** Given horizon length $H$ and the dataset sample size of $\mathcal{D}$ as $N_\mathcal{D}$, the behavior cloning performance bound of IDRL at iteration $k$ is given by

$$V(\pi) = V(\mathcal{D}_{k+1}) - \mathcal{O}\left(\frac{|\mathcal{S}|H^2}{N_{\mathcal{D}_{k+1}} + N_{\mathcal{D}_k - \mathcal{D}_{k+1}}/\max_s w_{k+1}^*(s)}\right).$$

*Proof.* The proof is highly built on the Theorem 3 in Li et al. (2024), which gives the performance bound of doing imitation learning on expert dataset $\mathcal{D}_E$ plus a supplementary dataset $\mathcal{D}_S$. If one uses the visitation distribution ratio $d^E(s,a)/d^S(s,a)$ to do weighted behavior cloning, the imitation gap bound is given by

$$\mathbb{E}\left[V(\pi^E) - V(\pi^{\text{ISW-BC}})\right] = \mathcal{O}\left(\frac{|S|H^2}{N_E + N_S/\mu}\right), \tag{14}$$

where $\mu = \max_{(s,h) \in \mathcal{S} \times [H]} \frac{d_h^{\pi^E}(s, \pi_h^E(s))}{d_h^{\pi^S}(s, \pi_h^E(s))}$. In our case, we can view IDRL at iteration $k$ as doing imitation learning on "expert" dataset $\mathcal{D}_{k+1}$ plus a supplementary dataset $\mathcal{D}_{k+1} - \mathcal{D}_k$, and we have the learned visitation distribution ratio $w_{k+1}^*(s) = d_{k+1}^*(s)/d_k^*(s)$. Put these results in we can get Theorem 3.

$\square$

**Theorem 4.** We have $V(\mathcal{D}_{k+1}) \geq V(\mathcal{D}_k)$ after the $k$-th iteration of IDRL.

*Proof.* Assuming the reward function is bounded, i.e, $r(s,a) \in [0, R_{max}]$. Note that $V(\mathcal{D}) = \mathbb{E}_{d^\mathcal{D}(s,a)}[r(s,a)]$ and because $d_{k+1}^*$ is the solution to

$$d_{k+1}^* = \arg\max_d \mathbb{E}_{d(s,a)}[r(s,a)] - \alpha D_f[d(s,a)\|d_k^*(s,a)],$$

so we have

$$\mathbb{E}_{d_{k+1}^*(s,a)}[r(s,a)] - \alpha D_f[d_{k+1}^*(s,a)\|d_k^*(s,a)] = \max_d \mathbb{E}_{d(s,a)}[r(s,a)] - \alpha D_f[d(s,a)\|d_k^*(s,a)]$$
$$\geq \mathbb{E}_{d_k^*(s,a)}[r(s,a)] - \alpha D_f[d_k^*(s,a)\|d_k^*(s,a)]$$
$$= \mathbb{E}_{d_k^*(s,a)}[r(s,a)].$$

Using these inequality and positivity of $f$-divergence, it follows that:

$$V(\mathcal{D}_{k+1}) = \mathbb{E}_{d_{k+1}^*(s,a)}[r(s,a)] \geq \mathbb{E}_{d_k^*(s,a)}[r(s,a)] + \alpha D_f[d_{k+1}^*(s,a)\|d_k^*(s,a)]$$
$$\geq \mathbb{E}_{d_k^*(s,a)}[r(s,a)] = V(\mathcal{D}_k)$$

$\square$

## C    EXPERIMENTAL DETAILS

For the Pearson $\chi^2$ we choose in practice, the corresponding $f, f^*, (f')^{-1}$ has the following form:

$$f(x) = (x-1)^2; \ f^*(y) = y(\frac{y}{4}+1); \ (f')^{-1}(R) = \frac{R}{2}+1$$

We apply one trick from Sikchi et al. (2023b) that rewrites objective (3) from $\alpha$ to $\lambda$ as

$$\min_V \mathbb{E}\left[(1-\lambda)V(s) + \lambda f_p^*(Q(s,a) - V(s))\right]$$

where $\lambda \in (0, 1)$ trades off linearly between the first term and the second term. This trick makes hyperparameter tuning easier as $\alpha$ has a nonlinear dependence through the non-linear function $f_p^*$.

**Toycase experimental details**    The dataset consists of 500 transitions collected via a discrete random policy, resulting in discretized state transitions. The weighted-BC policy, learned from filtered transitions, outputs continuous actions, illustrating the effect of weighted-BC on each transition. Both the policy and value networks are 3-layer MLPs with 256 hidden units. We used the Adam optimizer (Kingma & Ba, 2015) with a learning rate of $1 \times 10^{-4}$. The $\lambda$ was set to 0.6 to both 2 iterations, each running for $10^6$ steps. The first 500k steps of each iteration are dedicated to learning the action ratio, followed by the remaining steps to optimize for the state-action distribution ratio.

**D4RL datasets experimental details**    For all tasks, we conducted our algorithm for 2 iterations with $10^6$ steps for each iterations. First 500k steps of each iteration are dedicated to learn the action ratio, followed by the remaining steps to optimize for the state-action distribution ratio. In Mujoco locomotion tasks, we computed the average mean returns over 10 evaluations every $5 \cdot 10^4$ training steps, across 7 different seeds. For Antmaze and Kitchen tasks, we calculated the average over 50 evaluations every $5 \cdot 10^4$ training steps, also across 7 seeds. Following previous research, we standardized the returns by dividing the difference in returns between the best and worst trajectories in MuJoCo tasks. In AntMaze tasks, we subtracted 3 from the rewards to better fit with our algorithm.

For the network, we use 3-layer MLP with 256 hidden units and Adam optimizer (Kingma & Ba, 2015) with a learning rate of $1 \times 10^{-4}$ for both policy and value functions in all tasks. We also use a target network with soft update weight $5 \times 10^{-3}$ for $Q$-function.

We implemented IDRL using PyTorch and ran it on all datasets. We followed the same reporting methods as mentioned earlier. Baseline results for other methods were directly sourced from their respective papers. In IDRL, we have one parameters: $\lambda$. Because a larger $\lambda$ indicates a stronger ability to search for optimal actions, we select a larger $\lambda$ if the value of $V$ doesn't diverge. The values of $\lambda$ for all datasets are listed in Table 3.

Table 3: $\lambda$ used in IDRL

| Dataset | $\lambda$ |
|---|---|
| halfcheetah-medium-v2 | 0.5 |
| hopper-medium-v2 | 0.6 |
| walker2d-medium-v2 | 0.5 |
| halfcheetah-medium-replay-v2 | 0.6 |
| hopper-medium-replay-v2 | 0.6 |
| walker2d-medium-replay-v2 | 0.6 |
| halfcheetah-medium-expert-v2 | 0.5 |
| hopper-medium-expert-v2 | 0.5 |
| walker2d-medium-expert-v2 | 0.5 |
| antmaze-umaze-v2 | 0.6 |
| antmaze-umaze-diverse-v2 | 0.4 |
| antmaze-medium-play-v2 | 0.7 |
| antmaze-medium-diverse-v2 | 0.7 |
| antmaze-large-play-v2 | 0.7 |
| antmaze-large-diverse-v2 | 0.8 |

**Corrupted demonstrations experimental details** In this experiment setting, we introduce the noisy dataset by mixing the expert and random dataset at varying expert ratios using Mujoco locomotion datasets, thereby simulating dataset collected with most low-performing behavior policies. The number of total transitions of the noisy dataset is $1,000,000$. We provide details in Table 4.

Table 4: Noisy dataset of MuJoCo locomotion tasks with different expert ratios.

| Env | Expert ratio | Total transitions | Expert transitions | Random transitions |
|---|---|---|---|---|
| | 1% | 1,000,000 | 10,000 | 990,000 |
| Walker2d | 5% | 1,000,000 | 50,000 | 950,000 |
| | 10% | 1,000,000 | 100,000 | 900,000 |
| | 1% | 1,000,000 | 10,000 | 990,000 |
| Halfcheetah | 5% | 1,000,000 | 50,000 | 950,000 |
| | 10% | 1,000,000 | 100,000 | 900,000 |
| | 1% | 1,000,000 | 10,000 | 990,000 |
| Hopper | 5% | 1,000,000 | 50,000 | 950,000 |
| | 10% | 1,000,000 | 100,000 | 900,000 |

## D ETHICS STATEMENT

While IDRL has positive social impacts by helping to solve various practical data-driven decision making tasks, such as in robotics, healthcare, and industrial control, it is important to acknowledge that some potential negative impacts also exist. If crucial transitions in the dataset are filtered out, the trained model could behave unpredictably or even dangerously in certain situations.

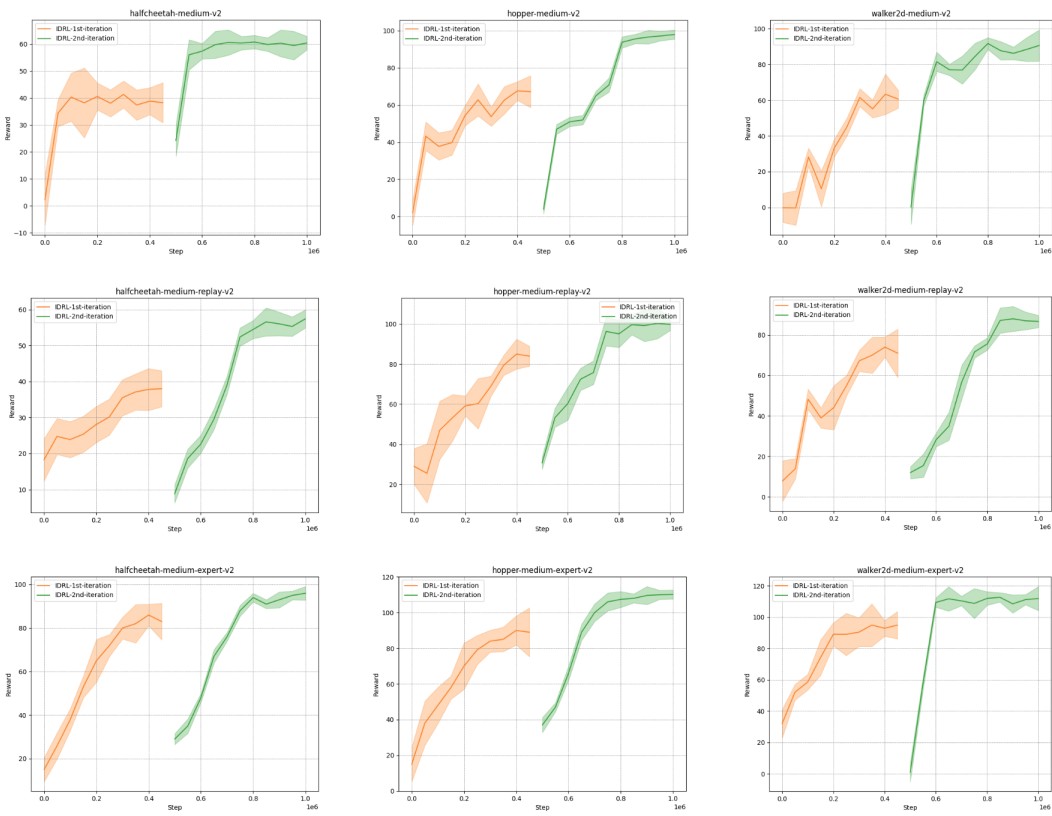

Figure 4: Learning curves of IDRL on D4RL Mujoco locomotion datasets.

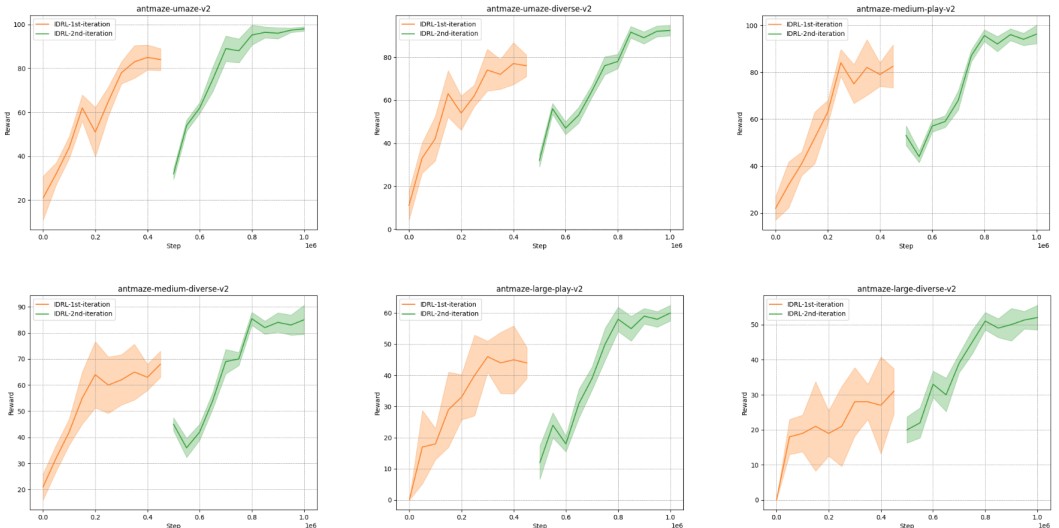

Figure 5: Learning curves of IDRL on D4RL Antmaze datasets.

