# OpenReview forum: "An Optimal Discriminator Weighted Imitation Perspective for Reinforcement Learning"
_ICLR.cc/2025/Conference — ICLR 2025 Poster_

### Official Review · Reviewer_msFL · 2024-10-24

**Soundness:** 3
**Presentation:** 3
**Contribution:** 3
**Rating:** 5
**Confidence:** 2

**Summary:**

This paper proposes Iterative Dual-RL (IDRL), a new algorithm for solving offline RL. The paper claims that an iterative "filtering weight" for imitation learning outperforms other offline RL methods. This point can be understood with the well-known dual formulation of RL, and by iterative self-distillation, the authors argue that RL can gradually correct state-action distribution between train and expert datasets. To validate this claim, the authors have provided theoretical justification for IDRL techniques and experimental results to support their claims.

**Strengths:**

1. The paper is well-written; the core contribution is straightforward to understand and supported by the theoretical arguments (I did not fully read the proof line-by-line).
2. This paper offers a novel perspective on offline RL. The authors successfully demonstrated that combining the dual formation of RL and imitation learning algorithms brings synergy to solve various tasks.
3. The paper contains realistic imitation learning experiments with corrupted datasets.

**Weaknesses:**

1. I believe the performance of IDRL is not stellar. Considering that the algorithm requires multiple dataset filtering steps, the performance gains from extra computation might not necessarily suggest the significance of the results.
2. Since it works with filtering, the algorithm might fail in scarcity of data. In MuJoCo, single demonstration imitation learning is a standard setting. In this case, I suspect IDRL's performance will converge with (single demonstration) behavior cloning performance.
3. An analysis scalability (such as computation costs) of various offline RL tasks should be reported and experimentally validated.

**Questions:**

1. Is the denoising process of diffusion-based offline RL (such as Diffusion-QL) similar to filtering datasets? How is IDRL conceptually different?

---

> ### Author Response · Authors · 2024-11-18
>
> We thank the reviewer for their thoughful review of our work. We address the reviewers questions and concerns below:
>
> >I believe the performance of IDRL is not stellar. Considering that the algorithm requires multiple dataset filtering steps, the performance gains from extra computation might not necessarily suggest the significance of the results.
>
> Note that on almost all tasks, IDRL matches or outperforms previous SOTA Primal-RL methods and greatly outperforms all Dual-RL methods. We acknowledge the additional computational cost brought by IDRL, however, we want to highlight that the core contibution of the paper is providing **the first**, **optimal discriminator-weighted** imitation view of solving offline reinforcement learning problem and showing the promising of correctly using Dual-RL methods. These new findings could potentially lead a new trend in the RL community.
>
>
> >Since it works with filtering, the algorithm might fail in scarcity of data. In MuJoCo, single demonstration imitation learning is a standard setting. In this case, I suspect IDRL's performance will converge with (single demonstration) behavior cloning performance.
>
> We respectfully disagree with this argument. Although IDRL leverages imitation, it does not operate at the trajectory level. Instead, IDRL could do stitching: extracting useful parts from different trajectories—those within the optimal distribution—effectively forming a superior demonstration compared to any single existing trajectory in the offline dataset. For example, in MuJoCo, Behavior Cloning (BC) on selected high-return trajectories in the dataset (as shown by the X%-BC results in the paper) performs significantly worse than IDRL.
>
>
> >An analysis scalability (such as computation costs) of various offline RL tasks should be reported and experimentally validated.
>
> We acknowledge the increased computational requirements of IDRL, as noted in the limitations section of our paper. To provide additional clarity, we present a runtime comparison of IDRL with several baseline offline RL algorithms below. On D4RL datasets, IDRL introduces only one additional iteration. During each iteration, we train $Q$, $V$ for half the training steps and $U$, $W$ for the other half. As a result, the total runtime is approximately twice that of IQL or SQL.
>
> |                 | IQL/SQL | CQL | Diffusion-QL | IDRL
> | --------------- | ----| ----| ----------   | ----
> |Run Time (training+evaluation) | ~5h| ~8h| ~8h    | ~11h
>
>
> >Is the denoising process of diffusion-based offline RL (such as Diffusion-QL) similar to filtering datasets? How is IDRL conceptually different?
>
> The success of diffusion-based methods relies on both accurate behavior modeling and effective sampling guidance. However, both steps can introduce errors. For instance, [1] demonstrates that diffusion behavior policies may produce potential out-of-distribution (OOD) actions, leading to overestimation errors in the guided sampling process.
>
> IDRL is related to these methods but differs significantly. IDRL is grounded in discriminator-weighted imitation learning, which avoids the use of potential OOD actions during training. This distinction ensures that IDRL does not inherit the overestimation errors associated with diffusion-based approaches.
>
> [1] [Diffusion-DICE: In-Sample Diffusion Guidance for Offline Reinforcement  Learning](https://arxiv.org/pdf/2407.20109)
>
> Please let us know if any further questions remain. We hope the reviewer can reassess our work with these clarifications.

---

> ### Comment · Reviewer_msFL · 2024-11-21
> **Response to Rebuttal**
>
> Dear authors, I thank you for your response; now I have a better understanding of IDRL's performance.
>
> I have a few questions on the theory.
> * Theorem 1 says that we need a decoupling approach from Eq. (5) s.t. $\omega^\ast(s,a) = \omega^\ast(s) \omega^\ast(a|s)$ based on Proposition 1. Is this interpretation correct?
> * Theorem 2 manifests an optimization problem for conditional visitation $\omega^\ast(a|s)$ from Theorem 1. Is this correct?
> * Theorem 3 is "highly built on the Theorem 3 in Li et al. (2024)." What could be the technical novelty in this theorem proving?

---

> > ### Author Response · Authors · 2024-11-21
> >
> > We thank the reviewer for the reply, about your question on the theory, we provide the responses below:
> >
> > - Proposition 1 proves that current Dual-RL methods learn the action distribution ratio $w^*(a|s)$ rather than the correct state-action distribution ratio $w(s,a)$, and Theorem 1 provides a way to recover $w^*(s)$ from $w^*(a|s)$ and we leverage that to get the correct state-action distribution ratio by following $w^*(s,a) = w^*(s) w^*(a|s)$.
> >
> > - Theorem 2 doesn't manifest an optimization problem for $w^*(a|s)$. Instead, it addresses the biased estimator issue in Obj.(9) and offers an unbiased estimator as a solution.
> >
> > - We acknowledge that this theorem builds upon “Theorem 3 in Li et al. (2024).” However, our contribution lies in how we leverage it to derive a theoretical analysis (performance bound) for IDRL. This extension is non-trivial since our work focuses on a different setting (offline RL) compared to Li et al. (2024) (offline imitation learning with supplemental datasets). Moreover, we propose Theorem 4, showing how the iterative process in IDRL minimizes the performance bound, which is also novel.
> >
> > To recall, the contribution of this paper is introducing the first optimal discriminator-weighted imitation view of solving offline RL, we propose a new algorithm based on Dual-RL to implement this idea and give theoretical and empirical analysis to it.

---

> > ### Author Response · Authors · 2024-11-25
> >
> > Please let us know if you have any further questions as the discussion period is ending soon. We would appreciate it if the reviewer could reassess our work in light of the clarifications provided and with a deeper understanding of our contributions.

---

> ### Comment · Reviewer_msFL · 2024-12-03
>
> Dear authors,
>
> Thank you for your detailed response. Based on your response and the reviews from other reviewers, I would like to take some additional time to carefully reevaluate the submission and finalize my recommendation, and further discuss with the other reviewers.

---

### Official Review · Reviewer_DNBG · 2024-10-26

**Soundness:** 3
**Presentation:** 3
**Contribution:** 3
**Rating:** 6
**Confidence:** 4

**Summary:**

The authors point out that current Dual-RL methods incorrectly estimate the visitation distribution ratio. As a remedy, they propose a method to recover the true visitation distribution ratio by solving an OPE problem using Fenchel-Rockafellar duality. Additionally, they introduce a method to iteratively refine the offline dataset using the learned distribution ratio. They theoretically analyze the performance bound and the monotonic improvement property of the filtering procedure. The authors perform experiments on a gridworld toy case and the D4RL benchmarks to validate their claims.

**Strengths:**

1. This work theoretically demonstrates that semi-gradient Dual-RL only learns an action-distribution ratio, and derives a method for recovering the full state-action visitation ratio with tractable objectives.
2. The proposed iterative filtering procedure is supported by theoretical analysis and empirical evaluations.

**Weaknesses:**

1. There should be a comparison of compute costs (e.g., run time, memory usage), given the substantial amount of modifications introduced (e.g., additional updates and iterative dataset refinement).
2. The proof of Theorem 1 lacks clarity for readers not familiar with Fenchel-Rockafellar duality, as the authors have omitted some details (e.g., solving for $w^{*}(s)$). A more detailed explanation would be helpful.
3. Line 240 states that Deep RL algorithms are prone to overestimation errors caused by fragmented trajectories. And the authors claim that the proposed method avoids this issue (Line 448-449). However, this fragmentation effect does not seem to be supported by any theoretical/empirical analysis in the paper or in a previous work. Please cite relevant texts if any.

**Questions:**

1. Equation 12 shows that $w^{*}(s, a) = w^{*}(s) * w^{*}(a | s)$, which implies that state-action pairs filtered by $w^{*}(a | s)$ would also be filtered by $w^{*}(s, a)$. If $w^{*}(a | s)$ produces fragmented trajectories during dataset refinement, the trajectories produced using $w^{*}(s, a)$ will only be more fragmented. Also, from looking at Figure 2(e), it appears that using $w^{*}(s, a)$ produces incomplete trajectories as well. How does correcting the visitation distribution address the fragmented trajectory problem?
2. Line 240 states that Deep RL algorithms are prone to overestimation errors caused by fragmented trajectories. Is this conclusion based on a previous study? To the best of my knowledge, the "stitching" challenge (which is a task design factor of D4RL) requires offline RL algorithms to assemble sub-trajectories in order to solve a task [1].
3. (Line 263, 283) Which equation are you referring to? I assume it is Equation 9?
4. Does the "IDRL w/ $w^{*}(a | s)$" result in Table 2 apply the iterative refinement procedure? If so, does iterative refinement contribute negatively with $w^{*}(a |s )$? Without distribution correction, one might expect the algorithm to produce results similar to conventional Dual-RL methods (e.g., IQL). However, the average score in Table 2 seems to be significantly worse (56.8 vs. 77.8 of IQL on Mujoco). A more detailed ablation study may help.

[1] Fu, Justin, Aviral Kumar, Ofir Nachum, George Tucker, and Sergey Levine. 2020. “D4RL: Datasets for Deep Data-Driven Reinforcement Learning.” _arXiv Preprint arXiv:2004.07219_. [http://arxiv.org/abs/2004.07219](http://arxiv.org/abs/2004.07219).

---

> ### Author Response · Authors · 2024-11-18
>
> We thank the reviewer for their thoughful review of our work. We address the reviewers questions and concerns below:
>
> >There should be a comparison of compute costs (e.g., run time, memory usage), given the substantial amount of modifications introduced (e.g., additional updates and iterative dataset refinement).
>
> We acknowledge the increased computational requirements of IDRL, as noted in the limitations section of our paper. To provide additional clarity, we present a runtime comparison of IDRL with several baseline offline RL algorithms below. On D4RL datasets, IDRL introduces only one additional iteration. During each iteration, we train $Q$, $V$ for half the training steps and $U$, $W$ for the other half. As a result, the total runtime is approximately twice that of IQL or SQL.
>
> |                 | IQL/SQL | CQL | Diffusion-QL | IDRL
> | --------------- | ----| ----| ----------   | ----
> |Run Time (training+evaluation) | ~5h| ~8h| ~8h    | ~11h
>
>
> >How does correcting the visitation distribution address the fragmented trajectory problem?
>
> The reviewer is correct in noting that states filtered with $w^*(a|s)$ are also filtered out by $w^*(s, a)$. The key distinction is that $w^*(a|s)$ does not account for the dynamics of the environment, resulting in a much higher degree of incompleteness (trajectory-level incomplete). Theoretically, filtering with $w^*(s, a)$ guarantees a valid visitation distribution (trajectory-level complete), ensuring that there are no trajectories with missing transitions. As the reviewer points out in Fig. 2e, some fragmentation is observed empirically, but the amount is much smaller compared to using $w^*(a|s)$, and it does not lead to performance collapse during training.
>
>
> >Line 240 states that Deep RL algorithms are prone to overestimation errors caused by fragmented trajectories. Is this conclusion based on a previous study? To the best of my knowledge, the "stitching" challenge (which is a task design factor of D4RL) requires offline RL algorithms to assemble sub-trajectories in order to solve a task [1].
>
> We agree with the reviewer that offline RL algorithms can stitch together trajectories to solve a task. However, such algorithms are typically trained on complete trajectories with few or no missing transitions, ensuring that every transition (except the last) is supported by at least one subsequent transition and has a valid Bellman backup target. In our work, fragmented trajectories refer to unsupported transitions that lack a valid backup target due to dataset filtering.
>
> To support our claim that fragmented trajectories lead to divergence issues, we ran IQL on randomly selected **transitions** from the original complete dataset. Specifically, we randomly selected 10K transitions from medium and medium-replay datasets in the Hopper and Walker environments and observed the learned value functions of IQL. The results, available at [link 1](https://ibb.co/bNvpF2k), [link 2](https://ibb.co/R3kwqm6), [link 3](https://ibb.co/tDVRhWw), and [link 4](https://ibb.co/Ykt1ThM), show divergence or overestimation in these cases.
>
> >(Line 263, 283) Which equation are you referring to? I assume it is Equation 9?
>
> Thanks for pointing this out, we have corrected the reference to Equation 9.
>
> >Without distribution correction, one might expect the algorithm to produce results similar to conventional Dual-RL methods (e.g., IQL). However, the average score in Table 2 seems to be significantly worse (56.8 vs. 77.8 of IQL on Mujoco). A more detailed ablation study may help.
>
> The ablation study uses two iterations to isolate the effect of filtering with $w(s, a)$. As mentioned earlier, running the second iteration with transitions filtered by $w(a|s)$ tends to cause divergence and degrade performance. Results for running one iteration without distribution correction can be inferred from the performance of f-DVL in Dual-RL [1], which closely aligns with IQL:
>
> |                 | IQL | f-DVL | IDRL w/ w(a\|s) (f-DVL with M=2)
> | --------------- | ----| ----      | ----------
> |Mean Score (Mujoco) | 77.8 |   75.7    | 56.8
>
> [1] [Dual RL: Unification and New Methods for Reinforcement and Imitation  Learning](https://arxiv.org/pdf/2302.08560)
>
>
> >The proof of Theorem 1 lacks clarity for readers not familiar with Fenchel-Rockafellar duality, as the authors have omitted some details (e.g., solving for w(s)). A more detailed explanation would be helpful.
>
>
> Thank you for pointing this out. The solution for $w(s)$ can be easily derived by setting $d^D$ to $d^*$ in the first term of Obj.(9). This is valid based on Lemma 2, which holds true for $d^*$. Taking the derivative and setting it to zero provides the solution for $w(s)$.
>
>
> Please let us know if any further questions remain. We hope the reviewer can reassess our work with these clarifications.

---

> > ### Comment · Reviewer_DNBG · 2024-11-19
> > **Response after rebuttal**
> >
> > I greatly appreciate the authors' comprehensive response. They have effectively addressed my initial concerns and provided additional details that considerably improve the quality of the paper. As a result, I am increasing my score with expectations that the authors will supplement the additional details (e.g., evidence of divergence, math details) in the revised manuscript.

---

> > > ### Author Response · Authors · 2024-11-19
> > >
> > > Thanks for raising the score!

---

### Official Review · Reviewer_oApW · 2024-11-04

**Soundness:** 3
**Presentation:** 3
**Contribution:** 4
**Rating:** 6
**Confidence:** 3

**Summary:**

This paper presents IDRL (iterative Dual RL), an algorithm for dual reinforcement learning that aims to solve two issues in current dual RL methods -- the semi gradient update and data regularized policy extraction. IDRL is a method which iteratively refines the dataset based on a trained discriminator. The paper proves both a theoretical iterative update guarantee and empirically shows that this method has superior performance compared to primal RL and dual RL offline methods.

**Strengths:**

- This paper does a good job with outlining the main issues with current dual RL algorithms and provides a theoretically grounded solution.
- While the idea is simple, it is well explained and well founded.
- The proposed method also has strong empirical results.

**Weaknesses:**

- It is unclear whether this method will suffer from poor generalization to other states which may have been ignored during dataset filtering.
- Further, this method seems to be computationally more expensive compared to other methods. It would be nice if this was discussed.

**Questions:**

- How does this policy generalize to states that were filtered out?
- How does filtering the dataset in round $k$, change the approximation of previously removed s,a pairs in later rounds?

---

> ### Author Response · Authors · 2024-11-18
>
> We thank the reviewer for their time and effort in reviewing our paper and for the constructive comments. Below, we address the concerns in detail:
>
> >Further, this method seems to be computationally more expensive compared to other methods. It would be nice if this was discussed.
>
> We acknowledge the increased computational requirements of IDRL, as noted in the limitations section of our paper. To provide additional clarity, we present a runtime comparison of IDRL with several baseline offline RL algorithms below. On D4RL datasets, IDRL introduces only one additional iteration. During each iteration, we train $Q$, $V$ for half the training steps and $U$, $W$ for the other half. As a result, the total runtime is approximately twice that of IQL or SQL.
>
> |  Algorithm      | IQL/SQL | CQL | Diffusion-QL | IDRL
> | --------------- | ----| ----| ----------   | ----
> |Run Time (training+evaluation) | ~5h| ~8h| ~8h    | ~11h
>
> >How does filtering the dataset in round k, change the approximation of previously removed s,a pairs in later rounds?
>
> Transitions filtered at iteration $k$ are no longer used in subsequent iterations. These transitions do not belong to the optimal distribution and are therefore excluded from further approximations.
>
>
> >How does this policy generalize to states that were filtered out?
>
> Thank you for this insightful question. This issue is less significant if the initial state distribution during deployment remains similar to the training data, which is typically the case in standard scenarios. States filtered during training will not be visited by the policy because they lie outside the distribution of the optimal policy. However, as we noted in the limitations section, we acknowledge that generalization issues may arise when the initial state distribution during deployment deviates significantly from the offline data. Note that in this case offline RL algorithms that don't filter out dataset may also encounter issues like suboptimality of the dataset action if behavior regularization weight is strong, or overestimation caused by the value function if behavior regularization weight is weak.
>
> Please let us know if any further questions remain. We hope the reviewer can reassess our work with these clarifications

---

> > ### Comment · Reviewer_oApW · 2024-11-20
> > **response to rebuttal**
> >
> > Thank you for addressing my questions. I will keep my score. I still believe that policy generalization to filtered out states would be a problem that could be further explored in this work. Indeed, many environments are stochastic which can lead to polices being in states that were filtered out.

---

> > > ### Author Response · Authors · 2024-11-20
> > >
> > > Thank you for your reply. The stochasticity of the environment is not an issue. If the environment is stochastic, the training data will naturally reflect this. The filtering process ensures that **all optimal states visited by the optimal policy** are retained in the data, guaranteeing that the policy will not encounter states that were filtered out. In other words, IDRL preserves transitions within the optimal state-action distribution, regardless of whether the environment’s dynamics are deterministic or stochastic.

---

> > > > ### Comment · Reviewer_oApW · 2024-11-25
> > > > **response to rebuttal**
> > > >
> > > > Thank you clearing up my misunderstanding. I am still wondering about this question:
> > > >
> > > > > How does this policy generalize to states that were filtered out?
> > > >
> > > > Adding small experiments to show at least some general robustness to states outside of the normal trajectory would be helpful.

---

> > > > > ### Author Response · Authors · 2024-12-02
> > > > >
> > > > > We thank the reviewer for their helpful suggestion, which has greatly contributed to improving our paper.
> > > > >
> > > > > We would like to provide an explanation **from the perspective of trading off between correct generalization and broader generalization to unseen states**, which may offer a more insightful answer to your question. Our paper presents the (discriminator-weighted) imitation learning perspective on solving offline RL. Our main claim is that while using more data (i.e., without filtering states) may increase robustness to unseen states, it comes at the risk of incorrect generalization due to low-quality transitions. Conversely, IDRL uses less data, which might reduce robustness to unseen states, but ensures that the generalization is correct. This trade-off is explicitly reflected in the theoretical analysis of IDRL, where the method iteratively finds the optimal balance. The theoretical bound for IDRL is expressed as:
> > > > >
> > > > > $$
> > > > > V\left(\pi\right) = V\left(D\right) - \mathcal{O}\left(\frac{|\mathcal{S}| H^2}{N_{D}  + ... }\right).
> > > > > $$
> > > > >
> > > > > This bound illustrates that the performance depends on both the size and the quality of the filtered dataset.
> > > > >
> > > > > Due to time constraints, we conducted experiments on the toy case presented in the paper to empirically validate this claim. Specifically, we compared the policies learned by using  $w(a | s)$ (dualRL) and  $w(s, a)$ (IDRL) on dataset states. Results are here (https://ibb.co/4j67Vth). The results demonstrate that IDRL ensures correct generalization, which is more critical than achieving broader but incorrect generalization (dualRL). Importantly, these findings also extend to more complex settings, as evidenced by the experimental results in our paper. Since behavior cloning errors caused by function approximation often lead to encountering unseen states during testing, the superior performance of IDRL highlights its ability to achieve correct generalization.
> > > > >
> > > > > It is worth noting that other approaches based on weighted behavior cloning also rely on using only part of the dataset (e.g., assigning small or zero weights to certain transitions when training the policy). For instance:
> > > > >  - SQL (https://openreview.net/forum?id=ueYYgo2pSSU) derives a sparse learning objective in principle.
> > > > >  - IQL assigns near-zero weights when the advantage is negative.
> > > > >  - Another ICLR 2025 submission (https://openreview.net/forum?id=elTJBP7Fbv, although achieving high review scores (8866) while performing worse than IDRL), also assigns some transition weights to zero via a new objective.
> > > > >
> > > > > What differentiates IDRL from these algorithms is that IDRL provides the first imitation learning perspective to justify why filtering data is both useful and necessary.

---

### Official Review · Reviewer_DfDQ · 2024-11-04

**Soundness:** 4
**Presentation:** 3
**Contribution:** 3
**Rating:** 6
**Confidence:** 3

**Summary:**

The paper presents a new framework, Iterative Dual-RL (IDRL), which utilizes an optimal discriminator-weighted imitation approach to enhance offline reinforcement learning (RL). This method iteratively refines the dataset to approximate the optimal visitation distribution by filtering out suboptimal transitions, thus aiming to overcome limitations of previous Dual-RL methods. IDRL is evaluated on D4RL benchmarks and several corrupted datasets, showing promising improvements in stability and performance over existing offline RL methods.

**Strengths:**

1.	The motivation of this paper is interesting and meaningful, which tries to combine offline RL with expert datasets.
2.	The proposed IDRL offers a novel discriminator-weighted imitation view that extends Dual-RL to better handle offline datasets by iteratively optimizing the dataset.
3.	Detailed theoretical derivations and empirical validations make the methodology clear and support the proposed approach’s effectiveness.
4.	The empirical results show that IDRL outperforms Primal-RL and existing Dual-RL methods on various benchmarks, indicating IDRL’s superior policy performance and dataset filtering effectiveness.

**Weaknesses:**

1.	This paper misses some literature in RL trained with weighted loss, such as EDP, and QVPO [1, 2].

[1] Kang B, Ma X, Du C, et al. Efficient diffusion policies for offline reinforcement learning[J]. Advances in Neural Information Processing Systems, 2024, 36.

[2] Ding S, Hu K, Zhang Z, et al. Diffusion-based Reinforcement Learning via Q-weighted Variational Policy Optimization[J]. arXiv preprint arXiv:2405.16173, 2024.

**Questions:**

1.	The reviewer believes the author should provide more explanation on how the additional variance introduced by the training of the U and W networks affects the overall stability of the algorithm.
2.	In Algorithm 1, the reviewer wonders whether the W network is updated based on (12) rather than (10) on line 10?
3.	The reviewer is confused about the value of M used in the experiments, and considers further clarification is needed here.

---

> ### Author Response · Authors · 2024-11-18
>
> We thank the reviewer for the effort engaged in the review phase and the constructive comments. Regarding the concerns, we provide the detailed responses separately as follows.
>
> >This paper misses some literature in RL trained with weighted loss, such as EDP, and QVPO [1, 2].
>
> We appreciate the reviewer bringing this to our attention and will include references to these works in the revised version of the paper. However, we would like to emphasize that the primary contribution of our paper is introducing **the first**, **optimal discriminator-weighted** imitation view of solving offline reinforcement learning, which is distinct from the methodologies presented in the mentioned literature.
>
> >The reviewer believes the author should provide more explanation on how the additional variance introduced by the training of the U and W networks affects the overall stability of the algorithm.
>
> Thank you for raising this important point. Based on our empirical evaluations, which span 24 datasets across 7 environments, we did not observe any divergence or instability during the training of $U$ and $W$. Theoretically, the learning of $U$ and $W$ is expected to be stable because their objectives are convex, which under mild assumptions guarantees convergence to the optimal solution. Additionally, the training processes for $U$ and $W$ are independent of the training of $Q$ and $V$; this can be viewed as a second phase of learning akin to training another set of $Q$ and $V$, which is similarly stable.
>
> That said, we acknowledge that potential instabilities may arise in broader cases due to the function approximation setting, where even $Q$-learning does not have guaranteed convergence.
>
>
> >In Algorithm 1, the reviewer wonders whether the W network is updated based on (12) rather than (10) on line 10?
>
> We apologize for the typo in Algorithm 1 and thank the reviewer for pointing it out. The W network is updated based on (10), not (12).
>
>
> >The reviewer is confused about the value of M used in the experiments, and considers further clarification is needed here.
>
> Sorry for the confusion, $M$ is the iteration number used in IDRL, as mentioned in Algorithm 1, line 2. We use M=2 on D4RL datasets and M=3 on corrupted demonstrations, we have made it more clear in the revised version of the paper.
>
>
> Please let us know if any further questions remain. We hope the reviewer can reassess our work with these clarifications

---

> ### Author Response · Authors · 2024-12-02
>
> We sincerely thank the reviewer once again for their time and effort in providing valuable feedback. We respectfully note that many papers receive improved scores during the rebuttal period after addressing reviewers’ concerns. Since there appear to be no remaining issues across all reviews for this paper, and considering your assessment of its soundness as excellent and its contributions as novel, we kindly ask if you might consider revisiting your score to further support this work, if you find it appropriate. We understand that such a request may seem unusual, but given that the reviewer who gave a score of 5 has not responded, the current statistics place our paper in a highly borderline position. We truly appreciate your understanding and consideration.

---

> ### Comment · Reviewer_DfDQ · 2024-12-03
>
> Thanks for your reply. Although this work offers a novel discriminator-weighted imitation view, it looks a little bit difficult to realize and needs extra costs for U and W network. In that case, I keep my score.
>
> Besides, I suggest the authors add a practical implementation part and instantiate the convex function $f$ as some common metrics (e.g., KL divergence). This will make it more straightforward to realize IDRL.

---

> > ### Author Response · Authors · 2024-12-03
> >
> > We thank the reviewer for the suggestion, but note that at the time this is posted more changes are not allowed to the manuscript by ICLR guidelines. IDRL is not complicated to implement despite requiring two additional networks. We provide psedocode in the main paper and our implementation details are available in detail in Appendix C where we instantiate f with chi-square divergence.  We will add code snippets to the appendix to make the implementation more clear and release code when further modifications are allowed to the submission.

---

### Meta-Review · Area_Chair_K5X2 · 2024-12-23

**Metareview:**

Thank you for your submission to ICLR. This paper presents Iterative Dual-RL (IDRL), which builds on the work of Dual-RL while trying to mitigate two issues: (1) difficulty in accurately estimating the state-action visitation ratio, and (2) learning a regularized, rather than optimal, visitation distribution ratio. IDRL aims to instead iteratively filter out suboptimal data, and then perform imitation learning on the remaining data close enough to the optimal visitation distribution.

Reviewers agree that the problem setting is well motivated, the proposed IDRL algorithm is novel, and the paper is well-written and clearly presented. On the other hand, multiple reviewers also point out concerns involving the increased computational costs of the method, compared with baseline methods. Regardless, a majority of the reviewers’ concerns were sufficiently addressed during the rebuttal phase, and I therefore recommend this paper for acceptance.

**Additional Comments On Reviewer Discussion:**

During the rebuttal period, the authors answered all of the reviewers’ questions thoroughly, and also provided an additional runtime comparison to shed light on the computational cost of their method. Reviewers, on the whole, appreciated these updates and most felt their concerns were addressed, which led to an increase in scores.

---

### Decision · Program_Chairs · 2025-01-22

Accept (Poster)